# Just Y-Prediction: Enabling Historical Cumulative Inconsistency in Label Diffusion for Learning with Noisy Labels

**Senyu Hou** [1]  **Gaoxia Jiang** [1]  **Xinyi Zheng** [1]  **Yaqing Guo** [1]  **Shuna Liang** [1]  **Wenjian Wang** [2]

## Abstract

Label noise is pervasive in real-world datasets and significantly compromises model generalization, fueling extensive research into Learning with Noisy Labels (LNL). Most LNL methods focus on robust discriminative learning, while recent generative classifiers such as label diffusion models (LDMs) show superior robustness by modeling class posteriors. However, current LDMs predominantly rely on standard $\epsilon$-prediction, where Gaussian noise lacks explicit class semantics, limiting both optimization and inference under label noise environments. To address this issue, we propose **just $y$-p**rediction (JYP), a novel training paradigm that enables LDMs to directly characterize the label manifold and leverage explicit class-semantic guidance. Theoretically, we prove that JYP converges to an optimal solution equivalent to that of $\epsilon$-prediction within the label diffusion framework, while facilitating accelerated convergence and enabling one-step inference. Leveraging JYP as a foundation, we further incorporate historical cumulative inconsistency to adaptively tailor optimization strategies for clean, noisy, and hard samples. Extensive experiments demonstrate that our method consistently outperforms competitors across diverse synthetic noisy datasets and achieves state-of-the-art performance on multiple real-world benchmarks.

## 1. Introduction

From deep neural networks (DNNs) to large language models (LLMs), advances across machine learning have in-

[1] School of Computer and Information Technology, Shanxi University, Taiyuan, Shanxi 030006, China [2] Key Laboratory of Data Intelligence and Cognitive Computing of Shanxi Province, Taiyuan, Shanxi 030006, China. Correspondence to: Wenjian Wang <wjwang@sxu.edu.cn>, Gaoxia Jiang <jianggaoxia@sxu.edu.cn>.

*Proceedings of the 43$^{rd}$ International Conference on Machine Learning*, Seoul, South Korea. PMLR 306, 2026. Copyright 2026 by the author(s).

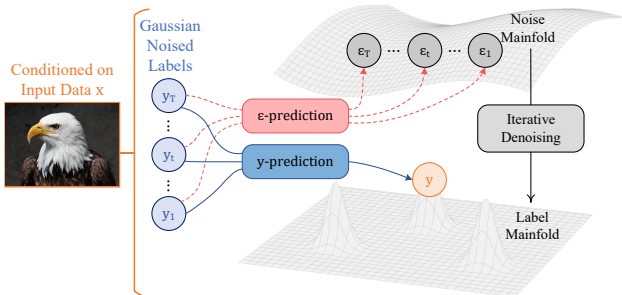

*Figure 1.* Illustration of our motivation. The $\epsilon$-prediction paradigm models noise on manifolds outside the label manifold, where the noise itself carries no semantic information. Each Gaussian noise sample corresponds to a distinct denoising direction, and the model relies on iterative denoising, preventing direct mapping from noisy observations back to the label space. The proposed JYP paradigm aims to ensure that all Gaussian noisy labels conditioned on $\mathbf{x}$ consistently map to the same origin, namely the ground-truth label $\mathbf{y}$. Due to the label manifold is low-entropy and strongly attractive, the model can always regress from multi-scale noisy observations toward the label manifold, maintaining meaningful inter-class relative differences in its predictions.

creasingly highlighted the need for large-scale, high-quality datasets (Chen et al., 2019). However, due to the high cost of manual and expert annotation, most real-world datasets rely on crowdsourcing or model-generated labels (Wang et al., 2022). This practice inevitably introduces label noise, compromising supervision reliability and presenting substantial challenges for model training.

Previous studies (Arpit et al., 2017) have demonstrated that noisy labels introduce bias into the optimization objective, resulting in degraded generalization performance. Therefore, developing effective learning methods that explicitly address label noise is crucial. Most existing learning with noisy labels (LNL) methods are based on discriminative classifiers and primarily focus on learning input-to-label mappings (Song et al., 2022). Under noisy conditions, these noise biases severely distort the modeling of class decision boundaries. Although robust techniques such as sample selection, label correction (Li et al., 2023), and semi-supervised learning (Li et al., 2020) can mitigate the impact of label noise, they do not address the overconfident predictions induced by discriminative modeling. Moreover, these

approaches often incur substantial computational overhead, and robust optimization struggles to overcome inherent performance bottlenecks.

With the rapid development of deep generative models, increasing attention has been devoted to their application in classification tasks (Yang et al., 2023). Generative classifiers (Zimmermann et al., 2021; Lee et al., 2019; Luo et al., 2025) model class posterior distributions, capturing both data characteristics and latent generative information. This model formulation enables them to generate smoother and more robust probability estimates under noisy conditions, making them particularly suitable for high-uncertainty scenarios such as label noise, missing information, and data scarcity (Chen et al., 2024; Belhasin et al., 2025). Label diffusion models (LDMs) represent a powerful class of generative classifiers and have demonstrated strong performance across a range of pattern recognition tasks, including image classification (Han et al., 2022; Belhasin et al., 2025), segmentation (Chen et al., 2023), and object detection (Li et al., 2025). More recently, they have also been successfully applied to LNL tasks (Chen et al., 2024; Hou et al., 2025). However, most existing LDMs are derived from denoising diffusion probabilistic models (DDPM) (Ho et al., 2020) and remain confined to the classical $\epsilon$-prediction paradigm. When applied to LNL tasks, this design introduces several fundamental limitations. **First**, during training, the $\epsilon$-prediction paradigm is constrained by the forward Gaussian transition kernel, forcing local noise predictions to lie on a high-dimensional manifold (as illustrated in Fig. 1). As a result, these Gaussian noise predictions lack explicit supervisory signals. This absence of direct supervision compels LDMs to rely on prior assumptions, such as purifying target label distributions or introducing additional regularization into the denoising objective. However, these approaches only mitigate the observable effects of label noise rather than addressing its root cause, and thus fail to fundamentally resolve the label noise issue. **Second**, the $\epsilon$-prediction paradigm inherently limits both the convergence speed of models and the accuracy of one-step inference, as shown in Fig. 2. These limitations lead to significant optimization challenges and further restrict the practical effectiveness of $\epsilon$-prediction LDMs in LNL tasks.

Inspired by recent work (Li et al., 2025; Li & He, 2025), we propose the just $y$-prediction (JYP) training paradigm for LDMs, theoretically establishing the equivalence between the two prediction paradigms and demonstrating the advantages of JYP within LDMs. Our motivation is illustrated in Fig. 1. Under the JYP paradigm, LDMs guided by image features recover a consistent $\mathbf{y}$ from noisy labels $\mathbf{y}_t$ at different timesteps. This serves as a multi-scale noise consistency constraint, enabling the model to learn mappings from arbitrary $\mathbf{y}_t$ to the low-dimensional label manifold $\mathbf{y}$, which is more direct and effective compared to $\epsilon$-prediction's it-

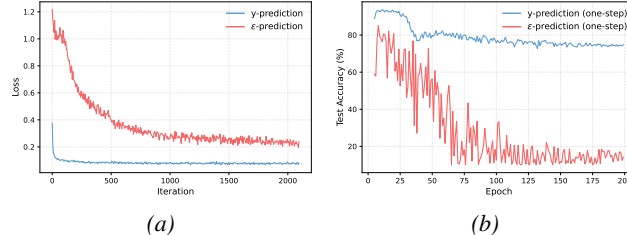

*Figure 2.* Comparison of convergence and one-step inference performance of different prediction paradigm. Results are reported on CIFAR-10 with 50% symmetric noise. (a) Training loss vs. iterations. (b) One-step inference accuracy vs. training epochs.

erative denoising from high-dimensional noise manifolds. As a result, label inference under the JYP paradigm can directly generate estimates close to the label manifold from high-noise inputs, achieving one-step inference capability. Similarly, JYP training occurs in the label space, enabling explicit supervision for selecting clean and noisy samples. However, during LDMs training, $\boldsymbol{y}$-predictions at each iteration remain affected by scaling effects induced by the noise level. To address this issue, we further integrate information via the historical cumulative inconsistency (HCI) of samples. We demonstrate that, under the JYP paradigm, the HCI offers a general-purpose mechanism for sample selection across various noise patterns, enabling the model to adaptively adjust the diffusion optimization objective based on the selected results.

The main contributions of this work are summarized as follows:

- We introduce a novel label diffusion training paradigm JYP, and theoretically establish its equivalence with the classical $\epsilon$-prediction paradigm, opening new avenues for LDMs applications.

- We analyze and empirically demonstrate that diffusion models trained under JYP achieve one-step inference, enabling end-to-end recovery of clean labels from random guess under feature guidance without iterative denoising.

- Leveraging JYP's ability to provide explicit class-related semantic information during training, we exploit HCI to extract generative information from labels across multiple noise scales for sample selection. Based on the selected results, we perform sample-level robust optimization on LDMs to improve generalization performance.

- Our method outperforms most state-of-the-art (SOTA) methods on both synthetic and real-world noisy datasets, and can seamlessly integrate with various pre-trained feature extractors for further performance improvements.

**Conflict of Interest Disclosure.** The authors declare that they have no financial conflicts of interest related to this work.

## 2. Related Work

**LNL.** Traditional LNL methods primarily rely on discriminative models that exploit performance discrepancies between clean and noisy samples for sample selection, often measured through loss magnitude (Han et al., 2018) and prediction confidence (Li et al., 2023). Following this, robust training strategies are employed, such as noise filtering (Han et al., 2018; Yu et al., 2019), label correction (Jiang et al., 2024; Xu et al., 2025), and semi-supervised learning (Li et al., 2020; Karim et al., 2022), to enhance model performance. Recent studies (Zhang et al., 2025a; Yuan et al., 2025) have shown that incorporating information accumulated throughout the training history can mitigate cognitive biases induced by incidental uncertainty during DNNs optimization, improving the alignment between sample selection mechanisms and robust training strategies. In contrast to prior discriminative approaches, our method focuses on generative classifiers, i.e., LDMs, and achieves robust optimization by leveraging generative information accumulated during training, rather than relying on conventional discriminative framework for LNL.

**Generative models for LNL.** Generative models are capable of modeling class-cluster distributions and rich generative information, and have been widely applied to both pattern recognition and LNL tasks. Some generative LNL approaches (Lee et al., 2019; Luo et al., 2025) aim to estimate unbiased class-cluster distributions by explicitly modeling data corrupted by label noise, and subsequently derive generative class posterior probabilities via Bayes theorem. Other studies (Bae et al., 2022; Chen et al., 2024; Hou et al., 2025) adopt a more direct and effective strategy by treating labels as generation targets and image features as conditioning variables, leveraging deep generative models for label correction or robust label generation. Among these approaches, LDM as a prominent framework, has been empirically validated as effective for LNL problems (Chen et al., 2024; Hou et al., 2025; Li et al., 2025). We investigate the limitations of conventional LDM paradigms in both training and inference, introducing a novel $y$-prediction paradigm that unlocks their potential to provide abundant generative class information for LNL.

## 3. Method: Just y-prediction for LDMs

### 3.1. Label Diffusion Models

Label diffusion models, an emerging class of generative classifiers, have been successfully applied to various pattern recognition and LNL tasks (Yang et al., 2023). These models outperform traditional discriminative models, usually using a framework derived from DDPM, where labels $\mathbf{y}$ serve as the diffusion object and features $\mathbf{x}$ act as the conditioning guidance. In the forward process, each step is controlled by a fixed Gaussian transition kernel:

$$q(\mathbf{y}_t \mid \mathbf{y}_{t-1}) := \mathcal{N}\left(\mathbf{y}_t; \sqrt{1-\beta_t^2}\mathbf{y}_{t-1}, \beta_t^2\mathbf{I}\right). \quad (1)$$

The sequence $\{\beta_t\}_{t=1}^T$ is a pre-determined noise schedule. Starting from an the initial label $\mathbf{y}_0$, the recursive application of the transition kernel yields a closed-form expression for the label distribution at time step $t$:

$$q(\mathbf{y}_t \mid \mathbf{y}_0) = \mathcal{N}\left(\mathbf{y}_t; \bar{\alpha}_t\mathbf{y}_0, (1-\bar{\alpha}_t^2)\mathbf{I}\right), \quad (2)$$

where $\bar{\alpha}_t := \prod_{t=1}^T \alpha_t = \prod_{t=1}^T \sqrt{1-\beta_t^2}$. This allows us to obtain $\mathbf{y}_t$ by adding noise to $\mathbf{y}_0$, independent of the feature $\mathbf{x}$, i.e., $\mathbf{y}_t = \bar{\alpha}_t\mathbf{y}_0 + \sqrt{1-\bar{\alpha}_t^2}\boldsymbol{\epsilon}$, $\boldsymbol{\epsilon} \sim \mathcal{N}(\mathbf{0}, \mathbf{I})$. The reverse transition kernel, based on the Markov property, is also Gaussian, with the following expression:

$$p(\mathbf{y}_{t-1} \mid \mathbf{y}_t, \mathbf{x}) = \mathcal{N}\left(\mathbf{y}_{t-1}; \boldsymbol{\mu}(\mathbf{y}_t, t, \mathbf{x}), \sigma_t^2\mathbf{I}\right), \quad (3)$$

where $\mu(\mathbf{y}_t, \mathbf{x}, t) := \frac{\bar{\alpha}_{t-1}\beta_t^2}{1-\bar{\alpha}_t^2}\mathbf{x} + \frac{(1-\bar{\alpha}_{t-1}^2)\alpha_t}{1-\bar{\alpha}_t^2}\mathbf{y}_t$ is the mean function, and $\sigma_t^2 := \frac{1-\bar{\alpha}_{t-1}^2}{1-\bar{\alpha}_t^2}\beta_t^2$ is the variance of the transition kernel. Typically, we parametrize the reverse distribution using a neural network as:

$$p_\theta(\mathbf{y}_{t-1} \mid \mathbf{y}_t, \mathbf{x}) := \mathcal{N}(\mathbf{y}_{t-1}; \boldsymbol{\mu}_\theta(\mathbf{y}_t, t, \mathbf{x}), \sigma_t^2\mathbf{I}). \quad (4)$$

The loss function for training is the squared error between the predicted mean and the true mean of the reverse transition distribution:

$$\mathcal{L}_{\text{LDM}}(\theta) = \sum_{t=1}^L \frac{1}{2\sigma_t^2} \|\boldsymbol{\mu}_\theta(\mathbf{y}_t, t) - \boldsymbol{\mu}(\mathbf{y}_t, \mathbf{y}_0, t)\|_2^2 + C. \quad (5)$$

For efficiency, mainstream LDMs (Han et al., 2022) often use an equivalent reparameterization approach $\boldsymbol{\mu}_\theta(\mathbf{y}_t, t) = \frac{1}{\alpha_t}\left(\mathbf{y}_t - \frac{1-\alpha_t^2}{\sqrt{1-\bar{\alpha}_t^2}}\boldsymbol{\epsilon}_\theta(\mathbf{y}_t, t)\right)$, known as $\boldsymbol{\epsilon}$-prediction. Substituting into the original loss equation results in the $\epsilon$-loss function, i.e., $\mathcal{L}_\epsilon = \|\boldsymbol{\epsilon}_\theta(\mathbf{y}_t, t) - \boldsymbol{\epsilon}\|_2^2$. Intuitively, the model no longer predicts the mean but instead estimates the random noise added at each step of the forward process. The predicted noise does not explicitly contain semantic information. Once trained, the model generates labels by denoising a randomly sampled initial state $\mathbf{y}_T \sim \mathcal{N}(\mathbf{0}, \mathbf{I})$ over $T$ steps:

$$\mathbf{y}_{t-1} = \frac{1}{\alpha_t}\left(\mathbf{y}_t - \frac{1-\alpha_t^2}{\sqrt{1-\bar{\alpha}_t^2}}\boldsymbol{\epsilon}_\theta(\mathbf{y}_t, t)\right) + \sigma_t \boldsymbol{z}_t, \quad (6)$$

where $\boldsymbol{z}_t \sim \mathcal{N}(\mathbf{0}, \mathbf{I})$ is used to control the randomness of the inference. Although this paradigm enables step-skipping sampling based on denoising diffusion implicit

*Table 1.* Two equivalent parameterizations for label diffusion training: $\mathbf{y}$-prediction and $\epsilon$-prediction. The linear transformations follow Theorem 3.1.

|  | **$y$-pred** $\mathbf{y}_\theta := \text{net}_\theta(\mathbf{y}_t, t, \mathbf{x})$ | **$\epsilon$-pred** $\epsilon_\theta := \text{net}_\theta(\mathbf{y}_t, t, \mathbf{x})$ |
|---|---|---|
| **$y$-loss** | $\mathbf{y}_\theta$ | $\mathbf{y}_\theta = \dfrac{\mathbf{y}_t - \sqrt{1-\alpha_t^2}\,\epsilon_\theta}{\alpha_t}$ |
| **$\epsilon$-loss** | $\epsilon_\theta = \dfrac{\mathbf{y}_t - \alpha_t\,\mathbf{y}_\theta}{\sqrt{1-\alpha_t^2}}$ | $\epsilon_\theta$ |

models (DDIM) (Song et al.), it still struggles to overcome the one-step inference bottleneck. Moreover, it fails to provide meaningful learnable information during training, which is unfavorable for solving the LNL problem.

### 3.2. Just y-prediction Training Scheme and Optimize Equivalence

To address the challenges faced by diffusion models based on the epsilon-prediction paradigm in training for the LNL problem, and inspired by related work on label progressive prediction (Li et al., 2025) and $x$-prediction in image diffusion (Li & He, 2025), we propose a training mechanism called just $\mathbf{y}$-prediction (JYP) for LDMs.

Reviewing the forward process of LDMs, the training objective is as shown in Eq. 5. Based on the closed-form mean of the reverse transition kernel, we introduce an alternative yet equivalent parameterization method called $\mathbf{y}$-prediction. In this approach, the neural network is trained to predict the target label $\mathbf{y}_\theta$ at a given time step $t$ from a noisy input $\mathbf{y}_t$. By replacing the ground-truth label $\mathbf{y}$ in the reverse mean expression with $\mathbf{y}_\theta(\mathbf{y}_t, t)$, we obtain the following model:

$$\boldsymbol{\mu}_\theta(\mathbf{y}_t, t) = \frac{\bar{\alpha}_{t-1}\beta_t^2}{1-\bar{\alpha}_t^2}\mathbf{y}_\theta(\mathbf{y}_t, t) + \frac{(1-\bar{\alpha}_{t-1}^2)\alpha_t}{1-\bar{\alpha}_t^2}\mathbf{y}_t. \quad (7)$$

Similar to the $\epsilon$-prediction, the training objective can be expressed as:

$$\|\boldsymbol{\mu}_\theta(\mathbf{y}_t, t) - \boldsymbol{\mu}(\mathbf{y}_t, \mathbf{y}_0, t)\|_2^2 \propto \|\mathbf{y}_\theta(\mathbf{y}_t, t) - \mathbf{y}_0\|_2^2, \quad (8)$$

where $\mathbf{y}_0 \sim p_{\text{label}}$ is considered the ground-truth label. The model is tasked with predicting the initial label, as close as possible to the ground truth, given the noisy observed label $\mathbf{y}_t$ at any given time step $t$. We refer to this loss function as the $\mathbf{y}$-loss ($\mathcal{L}_y$). Although the $\mathbf{y}$-prediction objective is formulated differently from the $\epsilon$-prediction paradigm commonly used in diffusion models, it raises a natural question as to whether optimizing the $\mathcal{L}_y$ fundamentally alters the denoising objective or the optimal solution of the model. To clarify the relationship between these two formulations, we next provide a theoretical analysis of the optimal predictors under the $\mathbf{y}$-prediction and $\epsilon$-prediction objectives.

Let $\{\mathbf{y}_t\}_{t\in[0,T]}$ denote a forward diffusion process defined by Eq. 2. Consider two parameterizations of the

reverse-process mean $\boldsymbol{\mu}_\theta(\mathbf{y}_t, t)$: (i) $\mathbf{y}$-prediction and (ii) $\epsilon$-prediction. The associated LDM objectives correspond to time-dependent weighted least-squares losses.

**Proposition 3.1 (Equivalence of $\mathbf{y}$-prediction and $\epsilon$-prediction).** *At the population level, the optimal solutions to the $\mathbf{y}$-prediction and $\epsilon$-prediction LDM objectives are given by*

$$\mathbf{y}^*(\mathbf{y}_t, t) = \mathbb{E}[\mathbf{y}_0 \mid \mathbf{y}_t, \mathbf{x}], \qquad \epsilon^*(\mathbf{y}_t, t) = \mathbb{E}[\epsilon_t \mid \mathbf{y}_t, \mathbf{x}].$$
$$(9)$$

*Moreover, these optimal solutions are related by invertible linear transformations. Specifically,*

$$\epsilon^*(\mathbf{y}_t, t, \mathbf{x}) = \frac{\mathbf{y}_t - \alpha_t\,\mathbf{y}^*(\mathbf{y}_t, t, \mathbf{x})}{\sqrt{1-\alpha_t^2}}, \quad (10)$$

*and conversely,*

$$\mathbf{y}^*(\mathbf{y}_t, t, \mathbf{x}) = \frac{\mathbf{y}_t - \sqrt{1-\alpha_t^2}\,\epsilon^*(\mathbf{y}_t, t, \mathbf{x})}{\alpha_t}. \quad (11)$$

*(See proof in Appendix A).*

As a consequence, the $\mathbf{y}$-prediction and $\epsilon$-prediction parameterizations are mathematically equivalent, in the sense that they induce the same optimal denoising solution for the reverse diffusion process.

Since the optimal solutions for the two prediction modes are linearly related under their respective loss constraints, we can naturally transform between loss modes for the same output mode of the network (either $\mathbf{y}$ or $\epsilon$). All possible combinations are shown in Table 1. In the ablation study (Section 4.3), we analyze the classification performance of four combinations. The "$\mathbf{y}$-prediction + $\mathbf{y}$-loss" combination is competitive with and outperforms the traditional "$\epsilon$-prediction + $\epsilon$-loss", as well as the other cross-combination paradigms. We also found that $\mathbf{y}$-prediction can directly employ the more suitable cross-entropy (CE) loss for classification tasks, since JYP essentially predicts the expectation of the ground truth labels. Focusing solely on the CE between the given and predicted labels provides a more direct and effective optimization path. More importantly, JYP aligns better with the LNL task. First, JYP provides explicit class information during each iteration, avoiding the entanglement with abstract information from Gaussian noise. This enables the diffusion model to make distinct optimization decisions for noisy and clean samples. Second, predictions from a model trained with JYP remain in the label space, so during inference, there is no need for iterative denoising to bring predictions closer to the label distribution, as in $\epsilon$-prediction. Instead, the target label estimate is gradually updated across progressively decreasing noise scales. Thanks to the low-dimensional effect of label distribution, even with one-step inference, we can obtain label estimates that carry class semantics. The detailed inference process is shown in Algorithm 1.

---

**Algorithm 1** JYP Inference

**Input:** Model $\text{net}_\theta$, data input $\mathbf{x}$, Gaussian transition kernel $q(\cdot)$, time steps $t_1 > t_2 > \cdots > t_{N_t}$, inference step $N_i$

**Output:** Generated label $\mathbf{y}_\theta$

1: Sample random noise $\mathbf{y}_T \sim \mathcal{N}(\mathbf{0}, \mathbf{I})$
2: $\mathbf{y}_\theta \leftarrow \text{net}_\theta(\mathbf{y}_T, T, \mathbf{x})$
3: **for** $n = 1$ **to** $N_i - 1$ **do**
4:      Sample $\mathbf{y}_{t_n} \sim q(\mathbf{y}_{t_n} \mid \mathbf{y}_\theta)$ by Eq. 2
5:      $\mathbf{y}_\theta \leftarrow \text{net}_\theta(\mathbf{y}_{t_n}, t_n, \mathbf{x})$
6: **end for**

---

## 3.3. Historical Cumulative Inconsistency

To address the challenge of sample selection in LDMs training with noisy labels, we propose a robust diffusion optimization strategy based on historical cumulative inconsistency (HCI). By leveraging the JYP diffusion training paradigm, which directly models clean label distributions, HCI enables effective identification of noisy samples.

In discriminative models, loss values or prediction confidence are commonly used for noisy sample detection, but these criteria become unreliable in traditional LDMs training since samples may be evaluated under different noise intensities at different time steps. Fortunately, under the JYP paradigm, the diffusion model directly predicts the clean label distribution at any time step $t$, providing a stable basis to measure sample inconsistency.

Let $(x_i, \tilde{y}_i)$ denote the $i$-th training sample, where $\tilde{y}_i$ represents a potentially noisy label. At time step $t$, the diffusion model outputs logits and the corresponding class generation probability are $\mathbf{o}_i^{(t)}$ and $\mathbf{p}_i^{(t)} = \text{softmax}(\mathbf{o}_i^{(t)})$, respectively. We can obtain the generation inconsistency of each sample by calculating the L1 norm distance between $\tilde{y}_i$ and $\mathbf{p}_i^{(t)}$, i.e., $||\mathbf{p}_i^{(t)} - \tilde{y}_i||$. To obtain a stable inconsistency measure, we focus on the generation probability $\mathbf{p}_{i,\tilde{y}_i}^{(t)}$ of the given label, avoiding interference from non-target classes influenced by Gaussian noise. We define the generative inconsistency at time step $t$ as:

$$d_i^{(t)} = \mathbf{1} - \mathbf{p}_{i,\tilde{y}_i}^{(t)}, \tag{12}$$

which measures the disagreement between the model prediction and the given label. Clean samples tend to have smaller $d_i^{(t)}$, while noisy-label samples typically yield larger inconsistency.

Although the above definition of inconsistency avoids cross-class interference, the varying noise intensity at different time steps $t$ still causes inconsistencies in the numerical scale of $d_i^{(t)}$. To further refine sample selection, we apply a time-dependent weight $\phi_t$ and accumulate the scaled

inconsistency across epochs $e$:

$$HCI_i^{(e)} = \sum_{\tau=1}^{e} \phi_t d_{i,\tau}^{(t)}, \tag{13}$$

where $\phi_t$ is designed to mitigate the imbalance in relative magnitudes caused by varying noise intensities. Figs. 3a, 3b, and 3c show that HCI progressively separates noisy and clean samples by smoothing training fluctuations across epochs, and this separation remains stable as training proceeds. As shown in Fig. 3d, a simple binary Gaussian mixture model (GMM) on HCI can stably distinguish noisy samples from clean ones.

## 3.4. Robust diffusion optimizing for JYP

Although HCI can separate clean and noisy samples in expectation, its estimates may still be unreliable under complex distributions or high noise rates, causing confirmation bias where hard or long-tail clean samples are mistaken as noisy (See Appendix Fig. 9). To further improve selection reliability, inspired by DISC (Li et al., 2023), we adopt a dual-view constrained sample selection scheme and apply robust optimization to different subsets. Specifically, we use weak/strong augmented views and maintain two HCI histories accordingly. After warm-up, we fit a binary GMM to each HCI distribution to obtain the noise posterior $\pi_i$, and then construct a high-confidence subset by intersecting the selections from both views.

Let $\mathcal{C}_v$ and $\mathcal{N}_v$ denote the clean/noisy sets selected by the GMM under $v \in \{w, s\}$, where $w$ and $s$ correspond to the weak and strong views, respectively. We take the intersection to form high-confidence sets $\mathcal{C} = \mathcal{C}_w \cap \mathcal{C}_s$ and $\mathcal{N} = \mathcal{N}_w \cap \mathcal{N}_s$, while the remaining samples are treated as hard samples set $\mathcal{H} = \mathcal{D} \setminus (\mathcal{C} \cup \mathcal{N})$. This dual-view agreement criterion reduces confirmation bias by selecting samples only when both views are consistent.

After partitioning the samples into $\mathcal{C}$, $\mathcal{N}$, and $\mathcal{H}$, we design different objectives for each subset. For high-confidence clean samples in $\mathcal{C}$, we directly optimize the diffusion model with

$$\mathcal{L}_{\text{clean}}^v(i) = \mathcal{L}_y\left(\mathbf{p}_{i,v}^{(t_v)}, \tilde{y}_{i,v}\right), \quad v \in \{w, s\}. \tag{14}$$

For noisy samples in $\mathcal{N}$, we avoid hard-label supervision and construct corrected soft labels $\bar{\mathbf{y}}_{i,v} = \pi_{i,v}\mathbf{p}_{i,v}^{(t_v)} + (1 - \pi_{i,v})\tilde{\mathbf{y}}_{i,v}$ using the noise posterior $\pi_{i,v}$: with the loss

$$\mathcal{L}_{\text{noisy}}^v(i) = \mathcal{L}_y\left(\mathbf{p}_{i,v}^{(t_v)}, \bar{y}_{i,v}\right), \quad v \in \{w, s\}. \tag{15}$$

For hard samples in $\mathcal{H}$, where the dual views cannot reach a consistent judgment, directly using label supervision may amplify errors and lead to overfitting. To address this issue,

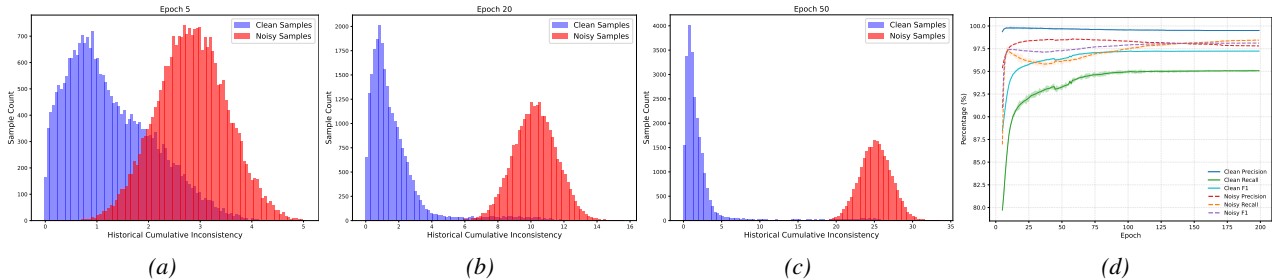

*Figure 3.* Distributions of HCI and the evolution of sample selection performance during training on CIFAR-10 with 50% symmetric noise. (a)-(c) Distributions at epochs 5, 20, and 50. (d) Performance curves of CP, CR, NP, NR, CF1, and NF1 over epochs 0–200.

we adopt a more robust generalized cross-entropy (GCE) to learning hard samples:

$$\mathcal{L}_{\text{hard}}(i) = \frac{1 - (\mathbf{p}_{i,w}^{(t_w)})^q}{q} + \frac{1 - (\mathbf{p}_{i,s}^{(t_s)})^q}{q}, \quad (16)$$

where $q \in (0, 1]$. Following (Zhang & Sabuncu, 2018), we also set $q = 0.7$. Besides the aforementioned subset-based supervision, we incorporate a dual-view consistency regularization term $\mathcal{L}_{\text{cons}}(i) = \left\| \mathbf{o}_{i,w}^{(t_w)} - \mathbf{o}_{i,s}^{(t_s)} \right\|_2^2$ and combine losses from two views to form the overall JYP loss:

$$\mathcal{L}_{\text{JYP}} = \mathbb{E}_{i \sim \mathcal{D}} \left[ (1 - \beta)\mathcal{L}^w(i) + \beta\mathcal{L}^s(i) + \mathcal{L}_{\text{cons}}(i) \right], \quad (17)$$

where $\beta \in [0, 1]$ is the view fusion coefficient. The detailed training process and the network architecture used are provided in Algorithm 2 and the Appendix Fig 7. Steps 1–4 are the initialization and warm-up phase, where in Step 2, we use a pre-trained model and simple neighbor estimation to pre-correct the samples for each view, which is also recommended by prior works (Chen et al., 2024; Hou et al., 2025). Steps 6–13 involve diffusion training for each epoch and accumulation of the HCI values, while Steps 14–15 update the model parameters and re-partition the samples for the next epoch.

## 4. Experiments

### 4.1. Experimental Settings

**Synthetic noisy datasets.** We evaluate the noise robustness of JYP on CIFAR-10/100 datasets with synthetic noisy labels of different types and rates. Let $T \in [0, 1]^{C \times C}$ be the noise transition matrix, where $T_{ij} = p(\tilde{y} = j \mid y = i)$ represents the probability that a sample with the true label $i$ is mislabeled as label $j$. We consider three widely used label noise settings: two class-conditional noise (CCN) models and one instance-dependent noise (IDN) model. **(i) Symmetric noise (Sym.):** labels are flipped uniformly to other classes with rate $r$, i.e., for $i \neq j, T_{ij} = \frac{r}{C-1}$. **(ii) Asymmetric noise (Asym.):** labels are flipped only between semantically similar classes according to a predefined $T$. **(iii)IDN:** the corruption depends on the input, i.e.,

**Algorithm 2** JYP Training

**Input:** Noisy training set $\mathcal{D} = \{(x_i, \tilde{y}_i)\}_{i=1}^N\}$, frozen pretrained model $f_p$, diffusion network $\text{net}_\theta$, number of epochs $E$, warm-up epochs $we$
**Output:** $\text{net}_\theta$
1: Obtain augmented features $\mathcal{D}^w$ and $\mathcal{D}^s$ using $f_p$.
2: Pre-correct labels for each view using KNN estimation
3: Initialize HCI for both views: $HCI_{i,w/s} \leftarrow 0$
4: Warm-up diffusion network $\text{net}_\theta$
5: **for** $e = we + 1$ **to** $E$ **do**
6:     **for** $v \in \{w, s\}$ **do**
7:         **for** each sample $(x_i, \tilde{y}_i)$ **in** $\mathcal{D}_v$ **do**
8:             Sample time slice $t_v \sim \{1, \ldots, T\}$ and Gaussian noise $\epsilon_v \sim \mathcal{N}(0, \mathbf{I})$
9:             Calculate $d_i^{(t_v)}$ using Eq. 12
10:             Update $HCI_{i,v}^{(e)} \leftarrow HCI_{i,v}^{(e-1)} + \phi_{t_v} d_i^{(t_v)}$
11:             Partition samples into $\mathcal{C}_v, \mathcal{N}_v$ using binary GMM based on HCI
12:         **end for**
13:     **end for**
14:     Re-partition samples based on the intersection of high-confidence sets from both views:
        $\mathcal{C} = \mathcal{C}_w \cap \mathcal{C}_s, \quad \mathcal{N} = \mathcal{N}_w \cap \mathcal{N}_s, \quad \mathcal{H} = \mathcal{D} \setminus (\mathcal{C} \cup \mathcal{N})$
15:     Take a gradient descent step on the loss (Eq. 17) to update diffusion network parameters
16: **end for**

$T_{ij} = p(\tilde{y} = j \mid y = i, x)$. Following standard settings, we use Sym. rates $r \in \{0.2, 0.5, 0.8\}$, Asym. rate $r = 0.4$, and IDN rates $r \in \{0.2, 0.4, 0.6\}$.

**Real-world noisy datasets.** We further evaluate JYP on four real-world noisy-label benchmarks: **(i) CIFAR-N (Wei et al., 2022):** a human-annotated noisy variant of CIFAR collected via Amazon Mechanical Turk. **(ii) Animal-10N (Song et al., 2019):** an animal classification dataset with naturally noisy labels. **(iii) Clothing1M (Xiao et al., 2015):** 1M clothing images from 14 categories, where labels are automatically extracted from online product descriptions with an estimated noise rate of 38.5%. **(iv) WebVision &**

*Table 2.* Overall performance on noisy CIFAR datasets. In the ResNet and ViT groups, the highest and second-highest accuracies are marked with bold and underline, respectively. Methods without any markings are taken directly from the original papers, and * indicate partial noise settings were reproduced by us, while ** indicates all results were reproduced by us.

| Methods | CIFAR-10 | | | | | | | | CIFAR-100 | | | | | | |
|---|---|---|---|---|---|---|---|---|---|---|---|---|---|---|---|
| | CCN-Sym. | | | CCN-Asym. | IDN | | | Human | CCN-Sym. | | | IDN | | | Human |
| | 20% | 50% | 80% | 40% | 20% | 40% | 60% | Worst | 20% | 50% | 80% | 20% | 40% | 60% | Fine |
| CE (Standard) | 86.80 | 76.40 | 52.90 | 72.00 | 83.93 | 67.64 | 43.83 | 77.69 | 62.00 | 43.70 | 19.90 | 57.35 | 43.17 | 24.42 | 55.50 |
| Co-teaching+ (ICML, 2019) | 89.50 | 85.70 | 67.40 | 76.91 | 89.80 | 73.78 | 59.22 | 83.26 | 65.60 | 51.80 | 27.90 | 41.71 | 24.45 | 14.58 | 57.88 |
| DivideMix (ICLR, 2020) | 96.10 | 94.60 | 93.20 | 93.40 | 93.33 | 95.07 | 85.50 | 92.56 | 77.30 | 74.60 | 60.20 | 79.04 | 76.08 | 46.72 | 71.13 |
| DISC (CVPR, 2023)* | 96.10 | 95.10 | 84.60 | 94.60 | 96.48 | 95.94 | 95.05 | 95.41 | 78.70 | 75.20 | 57.60 | 80.12 | 78.44 | 69.57 | 73.33 |
| DMLP (CVPR, 2023)* | 96.30 | 95.80 | 94.50 | 95.07 | 96.49 | 96.03 | 95.27 | 95.49 | 79.90 | 76.80 | 68.60 | 80.08 | 78.51 | 71.09 | 73.56 |
| RML (AAAI, 2024)* | 96.50 | 95.70 | 93.90 | 95.12 | 96.31 | 95.52 | 94.75 | 94.31 | 78.90 | 77.80 | 64.10 | 78.41 | 76.83 | 70.38 | 72.79 |
| JAL (ICCV, 2025)* | 89.99 | 85.03 | 64.84 | 79.54 | 89.90 | 86.78 | 75.02 | 81.25 | 68.33 | 61.59 | 23.11 | 67.77 | 63.56 | 51.69 | 59.38 |
| PLReMix (WACV, 2025)* | **96.63** | 95.71 | 95.08 | 95.11 | 95.91 | 95.36 | 94.08 | 95.03 | 77.95 | 77.78 | 68.76 | 77.95 | 77.67 | 68.41 | 72.98 |
| Early-Cutting (NeurIPS, 2025) | 93.79 | 91.70 | — | — | 93.40 | 90.78 | — | 87.43 | 76.20 | 72.27 | — | 75.03 | 69.94 | — | 66.52 |
| IDO (NeurIPS, 2025)** | 96.46 | 95.45 | 92.73 | 95.03 | 95.66 | 83.78 | 79.66 | 92.49 | **80.37** | **78.81** | 56.91 | **80.16** | 78.56 | 56.92 | 73.32 |
| JYP (Ours) | 96.36 | **96.07** | **95.48** | **95.78** | 96.49 | **96.08** | 95.42 | **95.67** | 78.27 | 77.96 | **70.70** | 79.75 | **78.59** | **71.50** | **73.87** |
| SIMIFEAT+ViT (ICML, 2022)** | 96.69 | 96.53 | 92.31 | 91.47 | 94.61 | 93.24 | 92.58 | 94.45 | 77.91 | 75.87 | 68.14 | 77.96 | 77.73 | 76.66 | 74.26 |
| EPL+ViT (ICLR, 2023)* | 96.24 | 96.10 | 96.00 | 94.10 | 95.47 | 95.90 | 95.11 | 94.57 | 79.83 | 76.50 | 72.30 | 78.03 | 77.80 | 77.34 | 74.32 |
| CLIPCleaner (MM, 2024)* | 95.92 | 95.67 | 95.04 | 94.89 | 95.51 | 95.48 | 95.02 | 95.35 | 78.21 | 75.23 | 69.72 | 76.37 | 74.98 | 72.90 | 73.21 |
| NoiseGPT (NeurIPS, 2024) | 96.20 | 94.90 | 92.60 | 92.80 | — | — | — | 96.90 | 76.30 | 71.50 | 63.90 | — | — | — | 73.24 |
| LRAD+ViT (NeurIPS, 2024)** | 96.68 | 96.34 | 85.29 | 95.68 | 96.48 | 96.44 | 95.56 | 95.71 | 78.62 | 77.07 | 68.90 | 78.09 | 77.87 | 74.53 | 74.14 |
| DLD+ViT (CVPR, 2025)* | 97.22 | 97.01 | 96.48 | 97.13 | 96.93 | 96.75 | 96.08 | 96.14 | 78.96 | 77.87 | 76.82 | 78.90 | 77.89 | 75.21 | 74.52 |
| PCL+ViT (ICML, 2025)** | 95.71 | 95.39 | 85.53 | 94.91 | 95.51 | 94.84 | 93.96 | 94.78 | 78.24 | 68.57 | 77.71 | 76.53 | 73.87 | 73.16 | |
| JYP+ViT (Ours) | **97.64** | **97.35** | **97.30** | **97.19** | **97.54** | **97.45** | **97.44** | **97.22** | **81.03** | **80.52** | **77.25** | **80.94** | **80.15** | **78.86** | **75.03** |
| | ±0.10 | ±0.15 | ±0.16 | ±0.09 | ±0.10 | ±0.12 | ±0.15 | ±0.09 | ±0.36 | ±0.41 | ±0.59 | ±0.47 | ±0.25 | ±0.31 | ±0.44 |

**ILSVRC12 (Li et al., 2017):** a webly supervised dataset covering 1,000 ImageNet (ILSVRC12) categories with 20% label noise; following standard practice, we evaluate on the first 50 categories of the Google Images subset.

**Baselines and implementation.** Previous studies suggest that pre-trained models improve noisy label learning stability and performance. Thus, we conduct experiments using two pre-trained networks: ResNet-50 and ViT-L/14. In the ResNet setup, we compare JYP with several representative LNL methods, including Co-teaching+ (Yu et al., 2019), DivideMix (Li et al., 2020), DISC (Li et al., 2023), DMLP (Tu et al., 2023), RML (Li et al., 2024), JAL (Wang et al., 2025), PLReMix (Liu et al., 2025), Early-Cutting (Yuan et al., 2025), and IDO (Zhang et al., 2025a). In the ViT setup, we select mainstream methods that also adapt pre-trained ViT models or other LLMs, including SIMIFEAT (Zhu et al., 2022), EPL (Ko et al., 2023), CLIPCleaner (Feng et al., 2024), NoiseGPT (Wang et al., 2024a), LRAD (Chen et al., 2024), DLD (Hou et al., 2025), and PCL (Li et al., 2025). On real-world noisy datasets, we introduce advanced methods such as Co-teaching (Han et al., 2018), RoG (Lee et al., 2019), MentorMix (Jiang et al., 2020), GJS (Englesson & Azizpour, 2021), UNICON (Karim et al., 2022), PCSE (Luo et al., 2025) for supplementary comparison. Detailed baseline introduction, evaluation metrics, training specifics, optimizer configurations, and data augmentation are provided in the Appendix D.

### 4.2. Results Analysis

**Overall performance on synthetic noisy datasets.** Table 2 shows that JYP achieves strong performance on CIFAR datasets with synthetic noise under both pre-trained model settings. When ResNet is used as the pre-trained backbone,

*Table 3.* Overall performance on real-world noisy datasets.

| Method | Animal-10N | Clothing1M | WebVision | | ILSVRC12 | |
|---|---|---|---|---|---|---|
| | | | Top-1 | Top-5 | Top-1 | Top-5 |
| Co-teaching (NeurIPS, 2018) | 84.86 | 69.21 | 63.58 | 85.20 | 61.48 | 84.70 |
| RoG (ICML, 2019) | 85.04 | 70.98 | 67.68 | — | 64.24 | — |
| DivideMix (ICLR, 2020) | 84.50 | 74.45 | 77.32 | 91.64 | 75.20 | 90.84 |
| MentorMix (ICML, 2020)* | 85.70 | 74.51 | 76.00 | 90.20 | 72.90 | 91.10 |
| GJS (NeurIPS, 2021) | 84.20 | 71.64 | 77.99 | 90.62 | 74.33 | 90.33 |
| UNICON (CVPR, 2022) | — | 74.98 | 77.60 | 93.44 | 75.29 | 93.72 |
| DISC (CVPR, 2023) | 87.10 | 74.79 | 80.28 | 92.28 | 77.44 | 92.28 |
| EPL (ICLR, 2023)* | 89.25 | 75.21 | 78.77 | 93.31 | 76.51 | 92.54 |
| CLIPCleaner (MM, 2024) | 88.90 | — | 81.56 | 93.26 | 77.80 | 92.08 |
| LRAD (NeurIPS, 2024)* | 88.60 | 75.70 | 84.16 | 95.07 | 82.56 | 94.63 |
| PCSE (TPAMI, 2025) | 85.48 | 71.37 | 70.48 | — | 67.72 | — |
| IDO (NeurIPS, 2025)* | 89.21 | 74.85 | 82.92 | 94.35 | 83.07 | 94.06 |
| DLD (CVPR, 2025) | 89.40 | 75.69 | 84.51 | 96.03 | 83.74 | 95.44 |
| PCL (ICML, 2025)** | 89.07 | 74.51 | 84.20 | 95.28 | 82.42 | 94.39 |
| JYP (Ours) | **90.40** | **75.77** | **84.62** | **97.60** | **85.08** | **98.20** |
| | ±0.31 | ±0.23 | ±0.40 | ±0.11 | ±0.24 | ±0.18 |

JYP outperforms competing methods under most noise configurations. In low-noise settings, the strict dual-view partition reduces the recall of clean samples, leading to overly strong regularization and a moderate performance drop compared with SOTA methods such as IDO. Nevertheless, JYP remains highly competitive. When ViT is adopted as the pre-trained model, JYP consistently surpasses all baselines that also leverage large-scale pre-training. For example, on CIFAR-100, JYP improves classification accuracy by 0.51% ∼ 3.65%, which indicates that, through generative modeling and robust optimization, JYP more effectively exploits representations provided by pre-trained models and remains robust across varying noise levels and patterns. Based on these findings, we recommend JYP+ViT as the default configuration. Additional experimental results on the CIFAR datasets are provided in the Appendix E.1.

**Overall performance on real-world noisy datasets.** Tables 2 and 3 show that JYP consistently outperforms competing methods on challenging real-world noisy datasets. On small-scale benchmarks such as Animal-10N, JYP exceeds

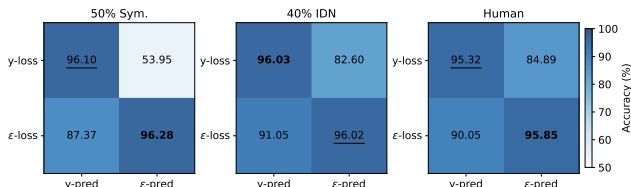

*Figure 4.* Classification performance of all combinations of prediction paradigms and loss functions (see Table 1) on the CIFAR-10 dataset under different noise settings.

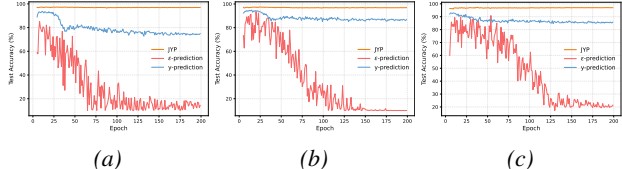

*Figure 5.* One-step inference performance on noisy CIFAR-10 using different LDM training methods. (a) 50% Sym. (b) 40% IDN. (c) *Worst* human noisy annotation.

*Table 4.* Ablation study on CIFAR-10 under different noise modes. Gap denotes the performance drop compared with the full model.

| Modules | | | Noise Mode | | | | | |
| --- | --- | --- | --- | --- | --- | --- | --- | --- |
| Pre-corr. | Dual-view | HCI | 50% Sym. | Gap↓ | 40% IDN | Gap↓ | Human | Gap↓ |
| ✓ | ✓ | ✓ | 97.35 | – | 97.45 | – | 97.22 | – |
| × | ✓ | ✓ | 97.01 | 0.34 | 97.26 | 0.19 | 96.89 | 0.33 |
| ✓ | × | ✓ | 96.58 | 0.77 | 96.66 | 0.79 | 96.60 | 0.62 |
| ✓ | × | × | 96.10 | 1.01 | 96.03 | 1.01 | 95.32 | 1.51 |
| × | × | × | 95.39 | 1.96 | 94.84 | 2.61 | 94.78 | 2.44 |

SOTA methods by over 1.3%, while on large-scale datasets like WebVision, JYP achieves comparable validation accuracy but demonstrates markedly superior generalization on ILSVRC12, even surpassing its own validation performance, in contrast to the limited generalization observed in other methods. These results confirm the effectiveness of JYP in enabling robust optimization of LDMs; additional results are provided in the Appendix E.2.

### 4.3. Ablation Studies

**$y$-prediction is critical.** As discussed in Table 1, $y$-prediction and $\epsilon$-prediction can be flexibly combined with the two loss formulations via linear transformations. The ablation results in Fig. 4 show that across diverse noise settings, models incorporating the $y$-prediction branch consistently achieve high and stable performance under different loss combinations. In contrast, when relying solely on $\epsilon$-prediction, model performance becomes highly sensitive to the choice of loss function and degrades substantially under the $y$-loss setting. This degradation is particularly pronounced in scenarios with heavy noise or more complex label distributions, because reconstructing $y$ from $\epsilon$ becomes increasingly ill-conditioned when $\alpha_t$ is small, thereby amplifying noise-prediction errors under sharp label-space losses (additional analysis is provided in the Appendix E.3). These results suggest that $\epsilon$-prediction alone is insufficient to capture the uncertainty structures introduced by noise during the label diffusion process. By contrast, $y$-prediction provides a direct and semantically aligned supervision signal, enabling the model to more effectively anchor true label semantics throughout the diffusion and reverse processes. In other words, regardless of the loss formulation, y-prediction fundamentally determines the upper bound of the model's performance, indicating that it is an essential component of the label diffusion paradigm.

**One-step inference performance of JYP.** As shown in Fig. 5, only $y$-prediction enables effective one-step inference, whereas $\epsilon$-prediction fails to do so. Although $\epsilon$-prediction exhibits limited one-step capability in the early training stage, its performance rapidly degrades and eventually collapses as the model learns the high-dimensional noise manifold. In contrast, $y$-prediction consistently main-

tains stable and usable one-step inference throughout training, indicating stronger alignment with the target label manifold. Building on this property, JYP further incorporates a HIC-based robust training strategy to refine the denoising trajectory from randomly corrupted labels toward clean labels, enabling consistently superior one-step inference accuracy across different noise settings.

**Effectiveness of the main modules.** Table 4 summarizes the ablation results under different noise patterns, where the complete JYP consistently achieves the best performance, demonstrating strong complementarity among modules. Removing the pre-correction module causes a slight drop, indicating its effectiveness in alleviating initial label noise, while removing the Dual-view or HCI module leads to more significant degradation. When both are removed, the model degenerates into a $y$-prediction variant of LRAD without effective sample selection or robust training, and further removing all modules reduces it to PCL with substantially worse performance across all noise settings. Overall, the proposed HCI contributes the most to performance improvement and serves as the core component for JYP. Pre-correction and Dual-view provide complementary benefits, supporting the method at the initialization stage and in improving sample selection accuracy, respectively. More ablation studies and analysis of $y$-prediction, JYP one-step inference, and hyperparameter sensitivity analysis are provided in the Appendix E.3

**Computational overhead analysis.** Fig. 6 reports the training and inference overhead of JYP. For training, JYP requires 24 seconds per epoch, which is slightly higher than LRAD (18s) and Co-teaching (20s), but remains substantially lower than DISC (36s), DivideMix (54s), and IDO (201s). This indicates that the added dual-view and HCI partitioning modules do not introduce a heavy computational

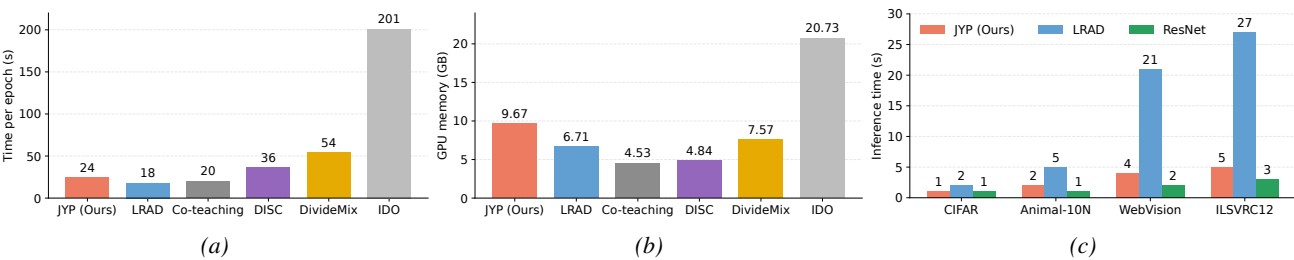

*Figure 6.* Computational overhead analysis. (a) Training time per epoch on CIFAR. (b) GPU memory per epoch on CIFAR. (c) Inference time on different test datasets. All statistics are measured with batch size 256 on a single NVIDIA GeForce RTX 4090 GPU.

*Table 5.* Classification accuracy (%) on UCI tabular datasets.

| Methods | CANE-9 | Spambase | Isolet | Letter | Australian | Blood | Breast Cancer | Credit | Titanic |
|---|---|---|---|---|---|---|---|---|---|
| Naive Bayes | 83.56 | 91.37 | 77.56 | 64.41 | 84.78 | 69.00 | 96.72 | 75.22 | 74.80 |
| SVM | 86.34 | 91.42 | 95.15 | 92.43 | 83.33 | 73.67 | 96.35 | 81.47 | 76.95 |
| Random Forest | 90.05 | **94.35** | 93.23 | 94.67 | **86.96** | 71.00 | **97.45** | 81.24 | 76.96 |
| XGBoost | 87.27 | 94.19 | 92.95 | 94.20 | 86.75 | 72.33 | **97.45** | 81.09 | 76.96 |
| LRAD | 92.59 | 93.91 | 95.87 | 94.61 | 86.23 | 77.00 | 97.08 | 81.82 | 78.09 |
| JYP (Ours) | **94.44** | 94.18 | **95.91** | **96.87** | 86.23 | **78.67** | **97.45** | **81.93** | **78.88** |

burden, since they are mainly used for sample selection rather than adding extra backpropagation-heavy networks. In terms of memory, JYP consumes 9.67 GB, which is higher than several lightweight baselines due to caching required by dual-view and HCI, but still far below IDO (20.73 GB). Therefore, the training overhead introduced by JYP is moderate and does not offset its robustness gains. For inference, JYP maintains clear efficiency advantages over LRAD across all test sets. On large-scale datasets, JYP reduces inference time from 21s to 4s on WebVision and from 27s to 5s on ILSVRC12, showing that its one-step inference is much more efficient than the iterative inference used by comparable label diffusion models. Meanwhile, JYP remains close to the standard ResNet classifier, indicating that its inference speed is comparable to discriminative models. These results show that JYP improves robustness without sacrificing practical training and inference efficiency.

**Extensibility of JYP on tabular data.** To further examine whether JYP can extend beyond image-based benchmarks, we evaluate a simplified JYP architecture on standard tabular-feature datasets. In this setting, following CARD's treatment of tabular data (Han et al., 2022), we remove the ResNet feature extractor used for image inputs and retain only a lightweight linear layer for feature-to-label prediction, while keeping the label diffusion training paradigm unchanged. The evaluated datasets cover heterogeneous input domains, including text-frequency features (CANE-9 and Spambase), speech features (Isolet), image-derived tabular features (Letter), and standard tabular classification benchmarks (Australian, Blood, Breast Cancer, Credit, and Titanic). We compare JYP with general machine learning baselines and LRAD.

As shown in Table 5, the simplified JYP variant remains effective across diverse tabular-feature domains. JYP achieves consistently competitive performance on UCI tabular datasets and outperforms LRAD on most evaluated datasets. Notably, on the text-feature CANE-9 dataset, JYP improves upon traditional machine learning baselines by 4.39%–10.88%, suggesting that the label diffusion formulation can also benefit non-visual feature spaces. These results indicate that the effectiveness of JYP does not depend on a heavy image backbone. Instead, by reducing the diffusion architecture to a lightweight linear feature-to-label predictor, JYP can naturally extend to tabular data while maintaining robust classification performance.

## 5. Conclusion

This paper proposes a novel label diffusion training paradigm for generative LNL, termed JYP, which overcomes the limitations of conventional LDMs based on $\epsilon$-prediction. By directly predicting clean labels at arbitrary noise scales, JYP introduces explicit class-semantic supervision, enabling faster convergence and effective one-step inference. Furthermore, JYP exploits historical accumulated inconsistency to mine generative information across multiple noise scales, guiding the diffusion model to perform targeted robust optimization through sample selection. Extensive experiments demonstrate that JYP significantly outperforms existing approaches on a wide range of synthetic and real-world noisy datasets. These findings highlight the strong potential of LDMs trained with JYP paradigm for LNL tasks and provide insights into extending diverse label diffusion optimization strategies to broader supervised learning studies.

## Acknowledgments

This work was supported in part by the National Natural Science Foundation of China under Grants U21A20513, 62576201, 62476157, 62276161, and 62406179, and in part by the Fundamental Research Program of Shanxi Province under Grant 202403021222026.

## Impact Statement

This paper aims to advance the field of Machine Learning by improving robustness to label noise through generative classification. By introducing the just $y$-prediction (JYP) paradigm for label diffusion models and leveraging historical cumulative inconsistency for sample-aware robust optimization, our work strengthens the long-term connection between generative modeling and robust learning, offering a principled way to utilize class-semantic signals and training history for more reliable supervision.

In the long run, we expect this line of research to benefit practitioners working with data-centric deep learning systems and large-scale models, where supervision quality is often imperfect due to crowdsourcing, web scraping, or automatic labeling. Our method provides a robust learning perspective that can mitigate the adverse effects of low-quality data and reduce the dependence on costly manual relabeling, potentially enabling more reliable deployment in diverse real-world settings.

This work does not propose new data collection pipelines, does not involve user-level profiling, and does not introduce capabilities that inherently increase misuse risk beyond standard supervised learning. We do not anticipate direct negative societal impacts arising from this research.

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

# Appendix

## A. Theoretical Analysis

**Proposition A.1** (**Equivalence of $y$-prediction and $\epsilon$-prediction**). *At the population level, the optimal solutions to the $y$-prediction and $\epsilon$-prediction LDM objectives are given by the conditional expectations:*

$$\mathbf{y}^*(\mathbf{y}_t, t) = \mathbb{E}[\mathbf{y}_0 \mid \mathbf{y}_t, \mathbf{x}], \qquad \boldsymbol{\epsilon}^*(\mathbf{y}_t, t) = \mathbb{E}[\boldsymbol{\epsilon}_t \mid \mathbf{y}_t, \mathbf{x}]. \tag{18}$$

*Moreover, these optimal solutions are related by invertible linear transformations. Specifically,*

$$\boldsymbol{\epsilon}^*(\mathbf{y}_t, t) = \frac{\mathbf{y}_t - \bar{\alpha}_t \, \mathbf{y}^*(\mathbf{y}_t, t)}{\sqrt{1 - \bar{\alpha}_t^2}}, \tag{19}$$

*and conversely,*

$$\mathbf{y}^*(\mathbf{y}_t, t) = \frac{\mathbf{y}_t - \sqrt{1 - \bar{\alpha}_t^2} \, \boldsymbol{\epsilon}^*(\mathbf{y}_t, t)}{\bar{\alpha}_t}. \tag{20}$$

*Proof.* To prove this proposition, we first establish a fundamental lemma regarding the minimizer of the squared loss, and then apply it to the specific loss functions of LDMs under different parameterizations.

**Lemma A.2** (**Optimal solution of least-squares regression**). *Let $(Y, Z)$ be any jointly distributed random variables with $Y \in \mathbb{R}^d$. Consider the function-space minimization problem:*

$$f^* \in \arg\min_f \mathbb{E}\big[\|f(Z) - Y\|_2^2\big]. \tag{21}$$

*Then the minimizer (almost surely) is the conditional expectation:*

$$f^*(Z) = \mathbb{E}[Y \mid Z]. \tag{22}$$

*Proof of Lemma A.2.* By the tower property of expectation, we can decompose the risk:

$$\mathbb{E}\big[\|f(Z) - Y\|_2^2\big] = \mathbb{E}_Z\Big[\mathbb{E}\big[\|f(Z) - Y\|_2^2 \mid Z\big]\Big]. \tag{23}$$

It suffices to minimize the inner conditional risk pointwise for each observation $Z = z$. Let $a = f(z)$ and $\mu(z) = \mathbb{E}[Y \mid Z = z]$. Expanding the squared norm:

$$\begin{aligned}
\mathbb{E}\big[\|a - Y\|_2^2 \mid Z = z\big] &= \mathbb{E}\big[\|(a - \mu(z)) + (\mu(z) - Y)\|_2^2 \mid Z = z\big] \\
&= \|a - \mu(z)\|_2^2 + \mathbb{E}\big[\|Y - \mu(z)\|_2^2 \mid Z = z\big] \\
&\quad + 2\langle a - \mu(z), \underbrace{\mathbb{E}[Y - \mu(z) \mid Z = z]}_{=0}\rangle.
\end{aligned} \tag{24}$$

The cross-term vanishes. The second term is independent of $a$. Thus, the expression is minimized when $\|a - \mu(z)\|_2^2 = 0$, implying $f^*(z) = \mu(z) = \mathbb{E}[Y \mid Z]$. $\diamond$

**Analysis of the LDM objective.** Recall that the LDM training objective is the mean-matching loss derived from the variational bound:

$$\mathcal{L}_{\mathrm{LDM}}(\theta) := \sum_{t=1}^L \frac{1}{2\sigma_t^2} \mathbb{E}_{\mathbf{y}_0, \mathbf{y}_t}\Big[\|\boldsymbol{\mu}_\theta(\mathbf{y}_t, t) - \boldsymbol{\mu}(\mathbf{y}_t, \mathbf{y}_0, t)\|_2^2\Big], \tag{25}$$

where the true posterior mean is $\mu(\mathbf{y}_t, \mathbf{y}_0, t) = \frac{\bar{\alpha}_{t-1}\beta_t^2}{1 - \bar{\alpha}_t^2}\mathbf{y}_0 + \frac{(1 - \bar{\alpha}_{t-1}^2)\alpha_t}{1 - \bar{\alpha}_t^2}\mathbf{y}_t$. The forward process provides the relation $\mathbf{y}_t = \bar{\alpha}_t \mathbf{y}_0 + \sqrt{1 - \bar{\alpha}_t^2}\,\boldsymbol{\epsilon}$.

**1. The case of $y$-prediction.** We parameterize the model mean to mimic the structure of the true mean as a linear interpolation of $\mathbf{y}_t$ and a predicted $\mathbf{y}_0$:

$$\boldsymbol{\mu}_\theta(\mathbf{y}_t, t) := a_t \, \mathbf{y}_\theta(\mathbf{y}_t, t) + b_t \, \mathbf{y}_t. \tag{26}$$

Substituting this into Eq. (25), the objective simplifies (up to constants and weighting $w_t$) to:

$$\mathcal{L}_y(\theta) \propto \sum_t w_t \, \mathbb{E}_{\mathbf{y}_0, \mathbf{y}_t} \left[ \|\mathbf{y}_\theta(\mathbf{y}_t, t) - \mathbf{y}_0\|_2^2 \right]. \tag{27}$$

For any fixed $t$, applying **Lemma A.2** with $Z = \mathbf{y}_t$ and $Y = \mathbf{y}_0$ immediately yields the optimal solution:

$$\mathbf{y}^*(\mathbf{y}_t, t) = \mathbb{E}[\mathbf{y}_0 \mid \mathbf{y}_t]. \tag{28}$$

**2. The case of $\epsilon$-prediction.** Alternatively, using the forward process relation $\mathbf{y}_0 = \frac{1}{\bar{\alpha}_t}(\mathbf{y}_t - \sqrt{1 - \bar{\alpha}_t^2} \epsilon)$, we can rewrite the true mean in terms of $\epsilon$. This motivates the parameterization:

$$\boldsymbol{\mu}_\theta(\mathbf{y}_t, t) := \frac{1}{\alpha_t}\big(\mathbf{y}_t - c_t \, \boldsymbol{\epsilon}_\theta(\mathbf{y}_t, t)\big). \tag{29}$$

Substituting this into Eq. (25), the objective simplifies to a weighted regression on the noise:

$$\mathcal{L}_\epsilon(\theta) \propto \sum_t \tilde{w}_t \, \mathbb{E}_{\mathbf{y}_0, \epsilon} \left[ \|\boldsymbol{\epsilon}_\theta(\mathbf{y}_t, t) - \boldsymbol{\epsilon}\|_2^2 \right]. \tag{30}$$

Again, applying **Lemma A.2** with $Z = \mathbf{y}_t$ and $Y = \boldsymbol{\epsilon}$, the optimal solution is:

$$\boldsymbol{\epsilon}^*(\mathbf{y}_t, t) = \mathbb{E}[\boldsymbol{\epsilon} \mid \mathbf{y}_t]. \tag{31}$$

**3. Equivalence and linear transformation.** Finally, we establish the relationship between Eq. (28) and Eq. (31). From the forward process $\mathbf{y}_t = \bar{\alpha}_t \mathbf{y}_0 + \sqrt{1 - \bar{\alpha}_t^2} \epsilon$, we can express $\epsilon$ as:

$$\boldsymbol{\epsilon} = \frac{\mathbf{y}_t - \bar{\alpha}_t \mathbf{y}_0}{\sqrt{1 - \bar{\alpha}_t^2}}. \tag{32}$$

Taking the conditional expectation $\mathbb{E}[\cdot \mid \mathbf{y}_t]$ on both sides:

$$\mathbb{E}[\boldsymbol{\epsilon} \mid \mathbf{y}_t] = \frac{\mathbf{y}_t - \bar{\alpha}_t \mathbb{E}[\mathbf{y}_0 \mid \mathbf{y}_t]}{\sqrt{1 - \bar{\alpha}_t^2}}. \tag{33}$$

Substituting the optimal solutions $\boldsymbol{\epsilon}^*$ and $\mathbf{y}^*$ derived above, we obtain:

$$\boldsymbol{\epsilon}^*(\mathbf{y}_t, t) = \frac{\mathbf{y}_t - \bar{\alpha}_t \mathbf{y}^*(\mathbf{y}_t, t)}{\sqrt{1 - \bar{\alpha}_t^2}}. \tag{34}$$

The inverse relationship follows by simple algebraic rearrangement. Thus, the optimal solutions are strictly equivalent. $\qquad\square$

**Explanation.** This equivalence follows from the fact that $L_{\text{LDM}}$ reduces to a squared-error regression problem under either parameterization, whose minimizer is universally the corresponding conditional expectation (as shown in Lemma A.2). Since the forward diffusion process establishes a deterministic linear relationship between the random variables $\mathbf{y}_0$, $\boldsymbol{\epsilon}$, and $\mathbf{y}_t$ (given any two, the third is determined), their conditional expectations given $\mathbf{y}_t$ must satisfy the same linear relationship.

## B. Details of the LDMs Implementation

### B.1. Unconditional Label Diffusion Models

In this section, we introduce unconditional LDMs following the theoretical framework of DDPM (Ho et al., 2020). We define the ground-truth label distribution $\mathbf{y}_0 \sim p_{\text{label}}$ and a Markovian forward process $q$, which progressively adds Gaussian noise $\epsilon \sim \mathcal{N}(0, \mathbf{I})$ to the labels based on diffusion strength $\beta_t$, diffusing label $\mathbf{y}_0$ to $\mathbf{y}_T$:

$$q(\mathbf{y}_t \mid \mathbf{y}_{t-1}) = \mathcal{N}\left(\mathbf{y}_t; \sqrt{1 - \beta_t^2} \mathbf{y}_{t-1}, \beta_t^2 \mathbf{I}\right). \tag{35}$$

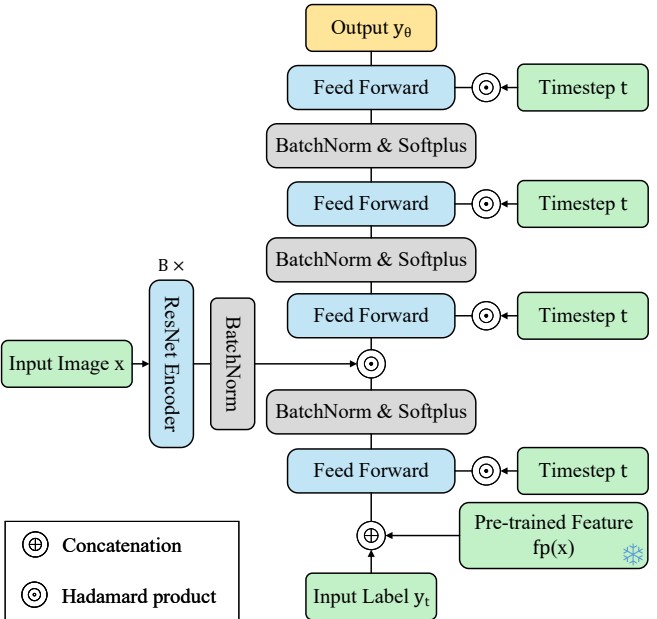

*Figure 7.* The network architecture for conditional label diffusion models. The input to the network consists of four elements: $\mathbf{y}_t$, $f_p(\mathbf{x})$, $\mathbf{x}$, and the time embedding for $t$, represented by green blocks. The blue blocks in the figure represent the trainable network components.

The joint distribution from $\mathbf{y}_0$ to $\mathbf{y}_t$ has a closed-form Gaussian expression:

$$q\left(\mathbf{y}_t \mid \mathbf{y}_0\right) = \mathcal{N}\left(\mathbf{y}_t; \bar{\alpha}_t \mathbf{y}_0, \left(1 - \bar{\alpha}_t^2\right) \mathbf{I}\right), \tag{36}$$

where $\bar{\alpha}_t := \prod_{t=1}^{T} \alpha_t = \prod_{t=1}^{T} \sqrt{1 - \beta_t^2}$. The posterior distribution of the forward process is:

$$p\left(\mathbf{y}_{t-1} \mid \mathbf{y}_t, t\right) = \mathcal{N}\left(\mathbf{y}_{t-1}; \boldsymbol{\mu}\left(\mathbf{y}_t, t\right), \sigma_t^2 \mathbf{I}\right). \tag{37}$$

For the reverse process, we employ a neural network to model the posterior $p\left(\mathbf{y}_{t-1} \mid \mathbf{y}_t, t\right)$, focusing on learning the mean:

$$p_\theta\left(\mathbf{y}_{t-1} \mid \mathbf{y}_t\right) = \mathcal{N}\left(\mathbf{y}_{t-1}; \boldsymbol{\mu}_\theta\left(\mathbf{y}_t, t\right), \tilde{\beta}_t \mathbf{I}\right). \tag{38}$$

The loss function to train LDM is simplified as:

$$\mathcal{L}_y := \mathbb{E}_{t \sim [1,T], \mathbf{y}_0 \sim q(\mathbf{y}_0), \epsilon \sim \mathcal{N}(0,\mathbf{I})} \left[\|\mathbf{y}_\theta\left(\mathbf{y}_t, t\right) - \mathbf{y}_0\|^2\right]. \tag{39}$$

This approach simplifies model training by directly predicting the ground-truth label $\mathbf{y}_0$. During inference, the model iteratively reconstructs the original label $\mathbf{y}_0$ from its noisy version $\mathbf{y}_t$ through a learned reverse sampling process. However, the unconditional label generation process serves as a foundational paradigm and does not incorporate any conditional information. This limitation makes it unsuitable for tasks like classification, where specific and distinct label assignments are essential.

### B.2. Conditional Label Diffusion Models and Network Architecture

While the unconditional label generation process, as discussed in the previous section, serves as a foundational paradigm, it fails to incorporate conditional information, which is crucial for tasks such as classification that require distinct label assignments. To address this limitation, we introduce the conditional version of LDMs, where features are incorporated as controlling conditions in the diffusion process. The key modification is to introduce a conditional Markov process $\hat{q}$, where the noise addition is the same as in the unconditional case, but the process now depends on additional features $\mathbf{x}$. Specifically, the forward process becomes:

$$\hat{q}\left(\mathbf{y}_t \mid \mathbf{y}_{t-1}, \mathbf{x}\right) := q\left(\mathbf{y}_t \mid \mathbf{y}_{t-1}\right), \tag{40}$$

---

**Algorithm 3** IDN generation algorithm

---

**Input:** Noisy set $\mathcal{N} = \{(x_i, y_i), \forall x_i \in \mathbb{R}^{S \times 1}\}$, noise ratio $r$
**Output:** Noisy instances $\{(x_i, \tilde{y}_i)\}_{i=1}^n$

1: Sample instance noise rate $q_i$ from the truncated normal distribution $\mathcal{N}(r, 0.1^2, [0, 1])$
2: Sample $W \in \mathbb{R}^{C \times S \times C}$ from the standard normal distribution $\mathcal{N}(0, 1^2)$
    // $C$ and $S$ indicate the number of classes and the dimension of instance features
3: **for** $i = 1, 2, ..., N$ **do**
4:     $p = x_i^T \times W_{y_i}$                                                    // Generate instance-dependent flip rates
5:     $p_{y_i} = -\infty$                                          // Control the diagonal entry of the instance-dependent transition matrix
6:     $p = q_i \times \text{softmax}(p)$                              // Make the sum of the off-diagonal entries of the $y_i$-th row to be $q_i$
7:     $p_{y_i} = 1 - q_i$
8:     Randomly choose a label from the label space according to the possibilities $p$ as noisy label $\tilde{y}_i$
9: **end for**

---

and the conditional posterior distribution becomes:

$$\hat{q}\left(\mathbf{y}_{t-1} \mid \mathbf{y}_t, \mathbf{x}\right) = \frac{\hat{q}\left(\mathbf{y}_{t-1}\right) \hat{q}\left(\mathbf{x} \mid \mathbf{y}_{t-1}\right)}{\hat{q}(\mathbf{x})}. \tag{41}$$

Using this conditional process, the label generation process is adapted to include both the noise and feature conditions. The posterior distribution for the reverse process is modified as:

$$\hat{q}\left(\mathbf{y}_{t-1} \mid \mathbf{y}_t, \mathbf{x}\right) \propto e^{-\|\mathbf{y}_{t-1} - \tilde{\mu}_t\|^2 / 2\tilde{\beta}_t} \cdot \hat{q}(\mathbf{x} \mid \mathbf{y}_{t-1}), \tag{42}$$

which ensures that feature conditions are properly incorporated into the label generation process. The key insight is that the network must be trained to learn the feature-conditional reverse diffusion process, which is achieved using a feature-guided architecture.

To implement this, we use the network architecture that combines a ResNet encoder for feature extraction with a series of feedforward layers to model the reverse process (See Fig. 7). The input to the network is a concatenation of the features $\mathbf{x}$ and the noisy labels $\mathbf{y}_t$, which are transformed using a forward process, then used as input to predict the noise term $\mathbf{y}_\theta$. However, incorporating $\hat{q}(\mathbf{x} \mid \mathbf{y}_{t-1})$ is challenging due to the often large dimensional gap between the features $\mathbf{x}$ and the labels $\mathbf{y}$. Drawing from previous work (Chen et al., 2024; Hou et al., 2025), we address this issue by introducing a pre-trained feature extractor $f_p$, which effectively bridges this dimensional gap. The extracted features are then concatenated with $\mathbf{y}_t$ before being input into the network, ensuring effective feature fusion. This feature-guided design allows the diffusion model to generate labels conditioned on input features, addressing the limitations of the unconditional diffusion model for classification tasks.

## C. More Detailed Baseline Description

This section reviews all baseline methods evaluated in the main experiments. These methods can be broadly categorized into four groups: sample selection, robust loss design, leveraging pre-trained models, and generative modeling.

**Sample selection.** Co-teaching (Han et al., 2018) and Co-teaching+ (Yu et al., 2019) are classical sample selection methods in LNL, which select training instances based on the small-loss criterion and improve robustness through dual-network co-training. MentorMix (Jiang et al., 2020) extends MentorNet (Jiang et al., 2018) by incorporating MixUp augmentation, demonstrating the effectiveness of semi-supervised learning paradigms for LNL. DivideMix (Li et al., 2020) partitions samples based on prediction confidence and adopts the semi-supervised strategy called MixMatch (Berthelot et al., 2019) to enhance training robustness. Building upon DivideMix, UNICON (Karim et al., 2022) further refines sample selection by introducing contrastive learning to improve selection accuracy. PLReMix (Liu et al., 2025) integrates contrastive loss with semantic information and model predictions to distinguish clean samples from noisy ones. DISC (Li et al., 2023), Early-cutting (Yuan et al., 2025), and IDO (Zhang et al., 2025a) enhance the stability of sample selection by exploiting historical confidence estimates or learning difficulty during training. Our method adopts a generative classification framework and also leverages historical information for sample selection, enabling robust optimization of LDMs.

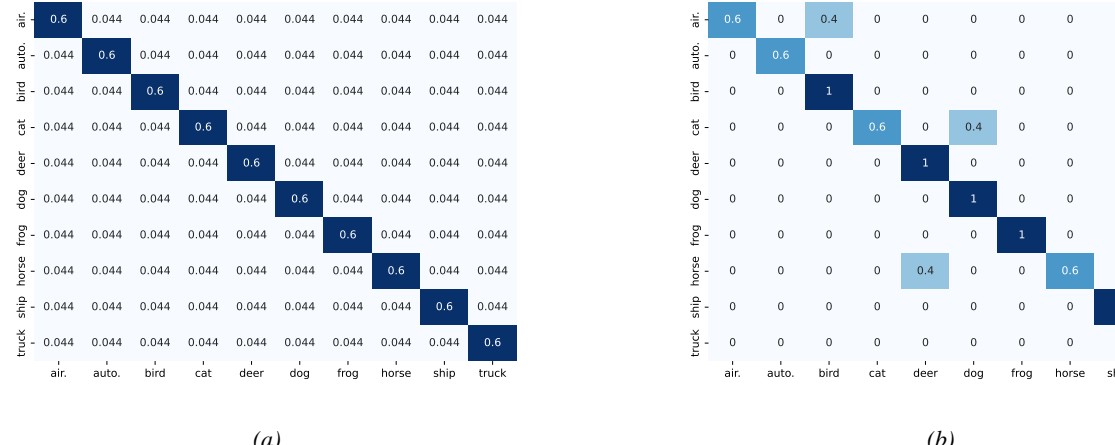

*(a)*                   *(b)*

*Figure 8.* Synthetic CCN transition matrices used in our experiments on CIFAR-10 with noise rate $r = 0.4$. (a) Symmetric noise. (b) Asymmetic noise.

**Robust loss functions.** GJS (Englesson & Azizpour, 2021) is a robust loss function derived from GCE (Zhang & Sabuncu, 2018) and Jensen–Shannon divergence, allowing smooth interpolation between cross-entropy and mean absolute error via a controllable mixing parameter. RML (Li et al., 2024) reconstructs loss values using aggregation strategies such as mean and median, yielding more robust loss estimates under label noise. JAL (Wang et al., 2025) is an asymmetric loss function based on active passive loss (APL) (Ma et al., 2020), which provides theoretical robustness guarantees. Our approach is also inspired by robust loss design and applies GCE loss to handle hard samples for enhanced robustness.

**Utilization of pre-trained models.** Prior studies (Zhu et al., 2022; Ko et al., 2023; Feng et al., 2024) have shown that properly leveraging pre-trained models can effectively mitigate LNL challenges. SIMIFEAT (Zhu et al., 2022) employs frozen pre-trained models for feature extraction and filters or corrects noisy labels based on neighborhood information. EPL (Ko et al., 2023) cleans noisy labels using pre-trained models and enhances training robustness by incorporating extracted features. CLIPCleaner (Feng et al., 2024) and NoiseGPT (Wang et al., 2024a) exploit powerful CLIP (Radford et al., 2021) and GPT (Achiam et al., 2023) models, respectively, to perform sample selection and label correction in a training-decoupled manner. Our method can also effectively integrate pretrained models to improve overall performance, demonstrating strong scalability.

**Generative models.** In LNL, mainstream generative approaches can be divided into two categories. The first category performs class-conditional modeling of data distributions and estimates posterior class probabilities via Bayes' rule, with representative methods including RoG (Lee et al., 2019) and PCSE (Luo et al., 2025). The second category treats labels as latent variables and conducts generative modeling using VAEs or diffusion models, such as NPC (Bae et al., 2022), LRAD (Chen et al., 2024), DLD (Hou et al., 2025), and PCL (Li et al., 2025). Our method is built upon LDMs and introduces improvements to both the training paradigm and robust optimization strategy, further extending the applicability of generative models to LNL.

## D. More Comprehensive Experimental Details

This section provides more comprehensive experimental details to facilitate reproducibility and deeper understanding of our evaluation protocol. Specifically, we describe the synthetic and real-world noisy datasets used in our experiments, the evaluation metrics for assessing both classification performance and sample selection quality, as well as the implementation details, including network architectures, training configurations, and data augmentation strategies.

### D.1. Datasets

**Synthetic noisy datasets.** We evaluate the noise robustness of JYP on the CIFAR-10 and CIFAR-100 benchmarks by injecting artificially synthesized label noise with different corruption mechanisms and noise rates. Specifically, let

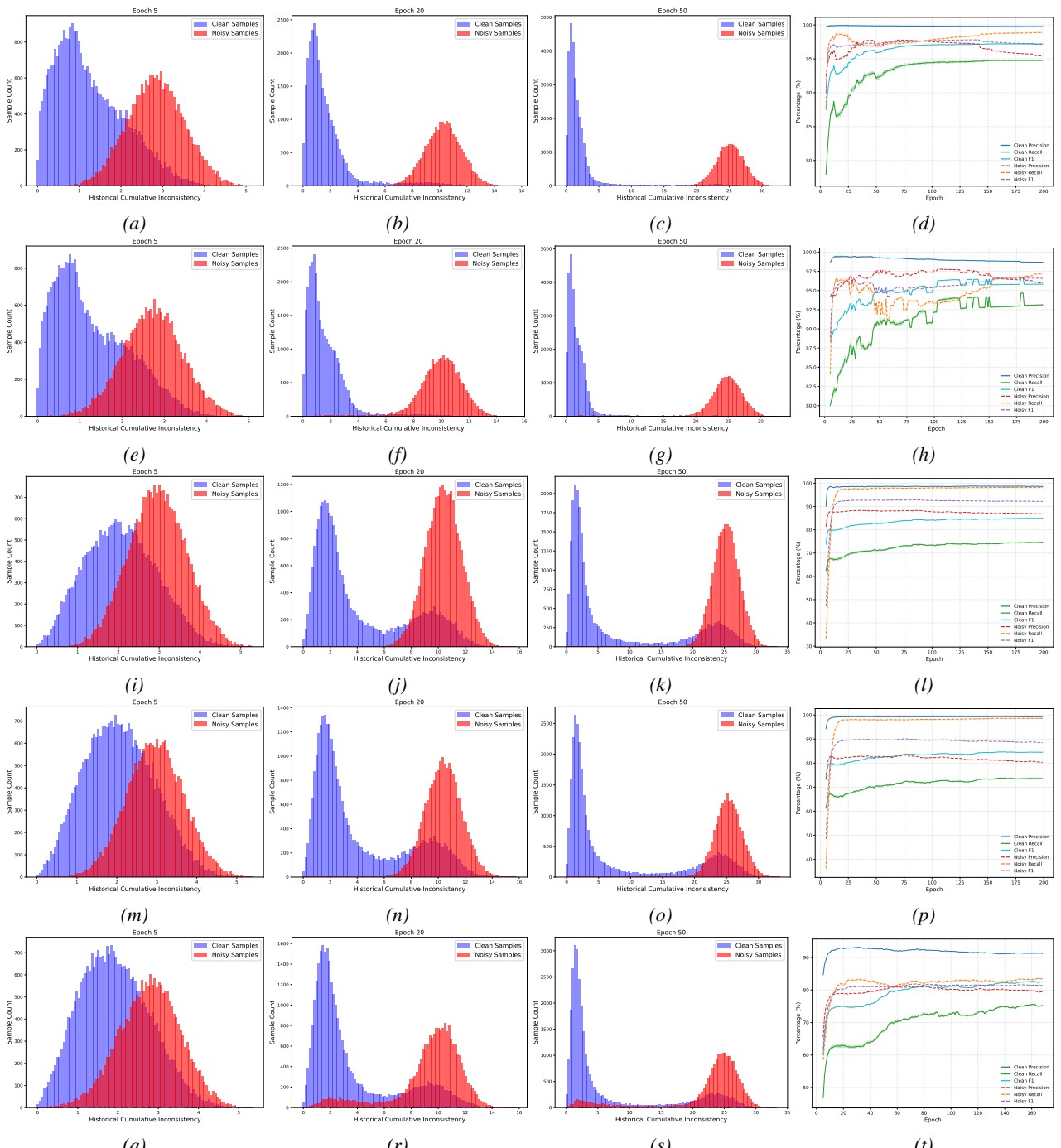

*Figure 9.* Distributions of historical cumulative inconsistency and the evolution of sample selection performance during training on CIFAR-10 and CIFAR-100 under synthetic and real-world noisy labels. (a)–(d) CIFAR-10 w/ 40% IDN. (e)–(h) CIFAR-10N w/ *Worst* human noisy annotation. (i)–(l) CIFAR-100 w/ 50% symmetric noise. (m)–(p) CIFAR-100 w/ 40% IDN. (q)–(t) CIFAR-100N w/ *Fine* human noisy annotation. The first three columns present the distributions at epochs 5, 20, and 50, and the fourth column reports the performance curves (CP, CR, NP, NR, CF1, and NF1) over epochs 0–200.

$T \in [0,1]^{C \times C}$ denote the noise transition matrix, where $T_{ij} = p(\tilde{y} = j \mid y = i)$ represents the probability that a sample with true label $i$ is corrupted into label $j$. Following common practice, we consider both class-conditional noise (CCN) and instance-dependent noise (IDN) to comprehensively assess robustness under diverse and realistic noise patterns. In this work, we adopt three widely used noise models.

*Table 6.* Overall performance on noisy CIFAR datasets with CCN. In the ResNet and ViT groups, our methods are highlighted with green and gray backgrounds, respectively. The highest and second-highest accuracies in each group are marked with bold and underlined text. Methods without any markings indicate the results are taken directly from the original papers, while * indicate the results were reproduced by us.

| Methods | CIFAR-10 w/ CCN | | | | CIFAR-100 w/ CCN | | |
|---|---|---|---|---|---|---|---|
| | Sym. 20% | Sym. 50% | Sym. 80% | Asym. 40% | Sym. 20% | Sym. 50% | Sym. 80% |
| CE (Standard) | 86.80 | 76.40 | 52.90 | 72.00 | 62.00 | 43.70 | 19.90 |
| Decoupling (Malach & Shalev-Shwartz, NeurIPS, 2017) | 87.40 | 79.57 | 51.08 | 78.67 | 62.75 | 47.59 | 17.38 |
| F-correction (Patrini et al., CVPR, 2017) | 86.80 | 79.80 | 63.30 | 87.20 | 61.50 | 46.60 | 19.90 |
| Mixup (Zhang et al., ICLR, 2018) | 90.17 | 79.94 | 57.15 | 82.68 | 63.65 | 46.94 | 14.11 |
| GCE (Zhang & Sabuncu, NeurIPS, 2018) | 90.05 | 79.40 | 50.67 | 74.73 | 59.92 | 50.22 | 18.53 |
| Co-teaching (Han et al., NeurIPS, 2018) | 87.95 | 78.06 | 57.48 | 81.14 | 66.03 | 55.33 | 15.18 |
| Co-teaching+ (Yu et al., ICML, 2019) | 89.50 | 85.70 | 67.40 | 76.91 | 65.60 | 51.80 | 27.90 |
| PENCIL (Yi & Wu, CVPR, 2019) | 92.40 | 89.10 | 77.50 | 88.50 | 69.40 | 57.50 | 31.10 |
| LossModelling (Arazo et al., ICML, 2019) | 94.00 | 92.00 | 86.80 | 87.40 | 73.90 | 66.10 | 48.20 |
| JoCoR (Wei et al., CVPR, 2020) | 94.46 | 93.33 | 81.31 | 90.98 | 74.70 | 66.45 | 55.50 |
| DivideMix (Li et al., ICLR, 2020) | 96.10 | 94.60 | 93.20 | 93.40 | 77.30 | 74.60 | 60.20 |
| ELR+ (Liu et al., NeurIPS, 2020,) | 95.80 | 93.80 | 93.30 | 92.30 | 77.60 | 73.60 | 60.80 |
| GJS (Englesson & Azizpour, NeurIPS, 2021) | 94.90 | 89.22 | 68.78 | 85.95 | 78.26 | 61.58 | 57.54 |
| UNICON (Karim et al., CVPR, 2022) | 96.00 | 95.60 | 93.90 | 94.10 | 78.90 | 77.60 | 63.90 |
| DISC (Li et al., CVPR, 2023) | 96.10 | 95.10 | 84.60 | 94.60 | 78.70 | 75.20 | 57.60 |
| DMLP (Tu et al., CVPR, 2023) | 96.30 | 95.80 | 94.50 | — | 79.90 | 76.80 | 68.60 |
| K-SPR (Wang et al., TPAMI, 2024) | 95.40 | — | 84.60 | 93.60 | 77.50 | — | 38.10 |
| HMW (Zhang et al., AAAI, 2024) | 93.50 | 95.20 | 93.70 | 93.70 | 76.60 | 75.80 | 63.40 |
| RML (Li et al., AAAI, 2024) | 96.50 | 95.70 | 93.90 | 95.12 | 78.90 | 77.80 | 64.10 |
| PCSE (Luo et al., TPAMI, 2025) | 91.92 | — | 71.35 | 89.03 | 72.46 | —— | 64.59 |
| JAL (Wang et al., ICCV, 2025) | 89.99 | 85.03 | 64.84 | 79.54 | 68.33 | 61.59 | 23.11 |
| Early-Cutting (Yuan et al., NeurIPS, 2025) | 93.79 | 91.70 | — | — | 76.20 | 72.27 | — |
| PLReMix (Liu et al., WACV, 2025) | **96.63** | 95.71 | 95.08 | 95.11 | 77.95 | 77.78 | 68.76 |
| DULC (Xu et al., AAAI, 2025) | 96.60 | 96.00 | 95.00 | 95.20 | 79.40 | 76.40 | 67.70 |
| IDO (Zhang et al., NeurIPS, 2025)* | 96.46 | 95.45 | 92.73 | 95.03 | **80.37** | **78.81** | 56.91 |
| JYP (Ours) | 96.36 | **96.07** | 95.48 | 95.78 | 78.27 | 77.96 | **70.70** |
| SIMIFEAT+ViT (Zhu et al., ICML, 2022)* | 96.69 | 96.53 | 92.31 | 91.47 | 77.91 | 75.87 | 68.14 |
| EPL+ViT (Ko et al., ICLR, 2023)* | 96.24 | 96.10 | 96.00 | 94.10 | 79.83 | 76.50 | 72.30 |
| CLIPCleaner (Feng et al., MM, 2024) | 95.92 | 95.67 | 95.04 | 94.89 | 78.21 | 75.23 | 69.72 |
| NoiseGPT (Wang et al., NeurIPS, 2024) | 96.20 | 94.90 | 92.60 | 92.80 | 76.30 | 71.50 | 63.90 |
| LRAD+ViT (Chen et al., NeurIPS, 2024)* | 96.68 | 96.34 | 85.29 | 95.68 | 78.62 | 77.07 | 68.90 |
| DLD+ViT (Hou et al., CVPR, 2025)* | 97.22 | 97.01 | 96.48 | 97.13 | 78.96 | 77.87 | 76.82 |
| PCL+ViT (Li et al., ICML, 2025)* | 95.71 | 95.39 | 85.53 | 94.91 | 77.69 | 76.24 | 68.57 |
| JYP+ViT (Ours) | **97.64 ± 0.10** | **97.35 ± 0.15** | **97.30 ± 0.16** | **97.19 ± 0.09** | **81.03 ± 0.36** | **80.52 ± 0.41** | **77.84 ± 0.59** |

**(i) Symmetric noise (Sym.):** each class label is independently flipped to any other class with equal probability. Concretely, given a noise rate $r \in [0, 1)$, for $i \neq j$ we set $T_{ij} = \frac{r}{C-1}$ and $T_{ii} = 1 - r$, which corresponds to uniformly random mislabeling across all incorrect classes.

**(ii) Asymmetric noise (Asym.):** this noise model captures more realistic annotation errors by restricting label corruption to semantically similar classes according to a predefined transition matrix $T$. For instance, on CIFAR-10, we flip the label 'cat' to 'dog', 'bird' to 'airplane', 'deer' to 'horse', and 'truck' to 'automobile' with probability $r$, while all other labels remain unchanged, leading to non-uniform and class-specific noise distributions. The noise distribution in CCN is fixed and defined by predetermined transition matrices, as illustrated in Fig. 8.

**(iii) Instance-dependent noise (IDN):** unlike CCN, the corruption process depends explicitly on the input sample $x$, such that $T_{ij} = p(\tilde{y} = j \mid y = i, x)$. Specifically, a noise rate is first sampled for each instance from a truncated normal distribution. Then, the label flip tendency toward each class is computed by projecting the sample features onto a randomly generated weight tensor. These tendencies are normalized via a softmax function to form an instance-dependent transition probability distribution. Finally, a noisy label is sampled from this distribution, replacing the original label. This process yields label noise that is both heterogeneous and feature-dependent. The full procedure is detailed in Algorithm 3. In this setting, each instance is assigned a distinct noise distribution determined by its features, resulting in heterogeneous and feature-aware label noise that is significantly more challenging to handle.

Following prior LNL studies (Li et al., 2023; Feng et al., 2024), we adopt symmetric noise rates $r \in \{0.2, 0.5, 0.8\}$, a fixed asymmetric noise rate of $r = 0.4$, and instance-dependent noise rates $r \in \{0.2, 0.4, 0.6\}$.

**Real-world noisy datasets.** To comprehensively evaluate the applicability and robustness of JYP under realistic label noise, we conduct experiments on four widely used real-world noisy datasets. These benchmarks span different data scales, noise sources, and annotation mechanisms, including human annotation noise, web-crawled weak labels, and large-scale

*Table 7.* Overall performance on noisy CIFAR datasets with IDN. In the ResNet and ViT groups, our methods are highlighted with green and gray backgrounds, respectively. The highest and second-highest accuracies in each group are marked with bold and underlined text. Methods without any markings indicate the results are taken directly from the original papers, while * indicate the results were reproduced by us.

| Methods | CIFAR-10 w/ IDN | | | CIFAR-100 w/ IDN | | |
|---|---|---|---|---|---|---|
| | 20% | 40% | 60% | 20% | 40% | 60% |
| CE (Standard) | $83.93 \pm 0.15$ | $67.64 \pm 0.26$ | $43.83 \pm 0.33$ | $57.35 \pm 0.08$ | $43.17 \pm 0.15$ | $24.42 \pm 0.16$ |
| Mixup (Zhang et al., ICLR, 2018) | $87.71 \pm 0.66$ | $82.65 \pm 0.38$ | $58.59 \pm 0.58$ | $46.31 \pm 0.25$ | $45.14 \pm 0.31$ | $23.77 \pm 0.26$ |
| GCE (Zhang & Sabuncu, NeurIPS, 2018) | $89.80 \pm 0.12$ | $78.95 \pm 0.15$ | $60.76 \pm 3.08$ | $58.01 \pm 0.26$ | $45.69 \pm 0.14$ | $35.08 \pm 0.23$ |
| Co-teaching (Han et al., NeurIPS, 2018) | $88.87 \pm 0.24$ | $73.00 \pm 1.24$ | $62.51 \pm 1.98$ | $43.30 \pm 0.39$ | $23.21 \pm 0.57$ | $12.58 \pm 0.58$ |
| Co-teaching+ (Yu et al., ICML, 2019) | $89.80 \pm 0.28$ | $73.78 \pm 1.39$ | $59.22 \pm 6.34$ | $41.71 \pm 0.78$ | $24.45 \pm 0.71$ | $12.58 \pm 0.58$ |
| DMI (Xu et al., NeurIPS, 2019) | $88.57 \pm 0.60$ | $82.82 \pm 1.49$ | $69.94 \pm 1.34$ | $57.90 \pm 1.21$ | $42.70 \pm 0.92$ | $26.96 \pm 2.08$ |
| Reweight-R (Xia et al., NeurIPS, 2019) | $90.04 \pm 0.46$ | $84.11 \pm 2.47$ | $72.18 \pm 2.47$ | $58.00 \pm 0.36$ | $43.83 \pm 8.42$ | $36.07 \pm 9.73$ |
| JoCoR (Wei et al., CVPR, 2020) | $88.78 \pm 0.15$ | $71.64 \pm 3.09$ | $63.46 \pm 1.58$ | $43.66 \pm 1.32$ | $23.95 \pm 0.44$ | $13.16 \pm 0.91$ |
| Peer Loss (Liu & Guo, ICML, 2020) | $89.12 \pm 0.76$ | $83.26 \pm 0.42$ | $74.53 \pm 1.22$ | $61.16 \pm 0.64$ | $47.23 \pm 1.23$ | $31.71 \pm 2.06$ |
| DivideMix (Li et al., ICLR, 2020) | $93.33 \pm 0.14$ | $95.07 \pm 0.11$ | $85.50 \pm 0.71$ | $79.04 \pm 0.25$ | $76.08 \pm 0.35$ | $46.72 \pm 1.32$ |
| CORSES2 (Cheng et al., ICLR, 2021) | $91.14 \pm 0.46$ | $83.67 \pm 1.29$ | $77.68 \pm 2.24$ | $66.47 \pm 0.45$ | $55.89 \pm 0.45$ | $38.55 \pm 3.25$ |
| CAL (Zhu et al., CVPR, 2021) | $92.01 \pm 0.75$ | $84.96 \pm 1.25$ | $79.82 \pm 2.56$ | $69.11 \pm 0.46$ | $63.17 \pm 1.49$ | $45.38 \pm 3.30$ |
| CC (Zhao et al., ECCV, 2022) | $93.68 \pm 0.12$ | $94.97 \pm 0.09$ | $94.95 \pm 0.11$ | $79.61 \pm 0.19$ | $76.58 \pm 0.25$ | $59.40 \pm 0.46$ |
| DISC (Li et al., CVPR, 2023) | $96.48 \pm 0.04$ | $\underline{95.94} \pm 0.04$ | $\underline{95.05} \pm 0.05$ | $\underline{80.12} \pm 0.13$ | $\underline{78.44} \pm 0.19$ | $\underline{69.57} \pm 0.14$ |
| RML (Li et al., AAAI, 2024)* | $96.31 \pm 0.21$ | $95.52 \pm 0.32$ | $94.75 \pm 0.27$ | $78.41 \pm 0.22$ | $76.83 \pm 0.49$ | $72.38 \pm 0.44$ |
| JAL (Wang et al., ICCV, 2025)* | $89.90 \pm 0.14$ | $86.78 \pm 0.17$ | $75.02 \pm 0.48$ | $67.77 \pm 0.38$ | $63.56 \pm 0.18$ | $51.69 \pm 0.59$ |
| Early-Cutting (Yuan et al., NeurIPS, 2025)* | $93.40 \pm 0.22$ | $90.78 \pm 0.31$ | $88.51 \pm 0.65$ | $75.03 \pm 0.23$ | $69.94 \pm 0.30$ | $61.06 \pm 0.39$ |
| PLReMix (Liu et al., WACV, 2025)* | $95.91 \pm 0.17$ | $95.36 \pm 0.45$ | $94.08 \pm 0.44$ | $77.95 \pm 0.17$ | $77.67 \pm 0.47$ | $68.41 \pm 0.35$ |
| DULC (Xu et al., AAAI, 2025)* | $\mathbf{96.57} \pm 0.23$ | $94.91 \pm 0.38$ | $94.07 \pm 0.25$ | $76.42 \pm 0.36$ | $\mathbf{78.57} \pm 0.29$ | $68.11 \pm 0.50$ |
| IDO (Zhang et al., NeurIPS, 2025)* | $95.66 \pm 0.21$ | $83.78 \pm 0.29$ | $79.66 \pm 0.48$ | $\mathbf{80.16} \pm 0.38$ | $78.56 \pm 0.36$ | $56.92 \pm 0.71$ |
| JYP (Ours) | $\underline{96.49} \pm 0.19$ | $\mathbf{96.08} \pm 0.13$ | $\mathbf{95.42} \pm 0.27$ | $79.75 \pm 0.26$ | $78.19 \pm 0.31$ | $\mathbf{71.50} \pm 0.28$ |
| SIMIFEAT+ViT (Zhu et al., ICML, 2022)* | $94.61 \pm 0.14$ | $93.24 \pm 0.33$ | $92.58 \pm 0.37$ | $77.96 \pm 0.29$ | $77.73 \pm 0.31$ | $76.66 \pm 0.52$ |
| EPL+ViT (Ko et al., ICLR, 2023)* | $95.47 \pm 0.23$ | $95.90 \pm 0.22$ | $95.11 \pm 0.34$ | $78.03 \pm 0.28$ | $77.80 \pm 0.29$ | $\underline{77.34} \pm 0.45$ |
| CLIPCleaner (Feng et al., MM, 2024)* | $95.51 \pm 0.16$ | $95.48 \pm 0.27$ | $95.02 \pm 0.41$ | $76.37 \pm 0.42$ | $74.98 \pm 0.29$ | $72.90 \pm 0.61$ |
| LRAD+ViT (Chen et al., NeurIPS, 2024)* | $\underline{96.48} \pm 0.25$ | $\underline{96.44} \pm 0.35$ | $\underline{95.56} \pm 0.40$ | $\underline{78.09} \pm 0.30$ | $\underline{77.87} \pm 0.34$ | $74.53 \pm 0.58$ |
| PCL+ViT (Li et al., ICML, 2025)* | $95.51 \pm 0.26$ | $94.84 \pm 0.27$ | $93.96 \pm 0.43$ | $77.71 \pm 0.32$ | $76.53 \pm 0.36$ | $73.87 \pm 0.63$ |
| JYP+ViT (Ours) | $\mathbf{97.54} \pm 0.10$ | $\mathbf{97.45} \pm 0.12$ | $\mathbf{97.44} \pm 0.15$ | $\mathbf{80.94} \pm 0.47$ | $\mathbf{80.15} \pm 0.25$ | $\mathbf{78.86} \pm 0.31$ |

instance-dependent noise, providing a thorough assessment of JYP across diverse LNL scenarios.

**(i) CIFAR-N** (Wei et al., 2022) is a real-world noisy-label benchmark derived from the CIFAR-10 and CIFAR-100 datasets, where the original training sets are re-annotated by human annotators. CIFAR-N consists of CIFAR-10N and CIFAR-100N, both containing 60,000 images with 50,000 for training and 10,000 for testing, while the test labels remain clean. For CIFAR-10N, each training image is independently labeled by three crowd workers on Amazon Mechanical Turk, resulting in multiple noisy-label settings, including *Aggregate* (majority voting), *Random 1/2/3* (single-annotator labels), and *Worst* (adversarially selected incorrect labels), with noise rates ranging from approximately 9% to over 40%. For CIFAR-100N, we adopt the *Fine* setting with human-annotated noisy labels, where the noise rate is about 40.2%. Overall, CIFAR-N captures instance-dependent, human-centric noise patterns and provides both noisy and clean labels for controlled evaluation of learning with real-world label noise.

**(ii) Animal-10N** (Song et al., 2019) contains 60,000 images, with 50,000 for training and 5,000 for testing. The images are sourced from Google and Bing, covering five visually similar animal class pairs (e.g., 'cat' vs. 'lynx'), resulting in approximately 8% noisy labels in the training set due to inter-class ambiguity. The test set is manually verified by experts to ensure label correctness.

**(iii) Clothing1M** (Xiao et al., 2015) is a large-scale real-world dataset for learning with noisy labels, containing approximately 1 million training images collected from online shopping websites. The labels are automatically generated from surrounding textual descriptions and thus suffer from substantial noise. In addition, Clothing1M provides a small clean subset of about 50K images with manually verified labels, as well as clean validation and test sets. The dataset covers 14 clothing categories, with an estimated label noise rate of around 38.5% in the noisy training set. Due to its large scale and severe, instance-dependent label noise, Clothing1M is widely used to evaluate the robustness and scalability of noisy-label learning methods.

**(iv) WebVision/ILSVRC12** (Li et al., 2017) contain a combined total of around 2.4 million images, collected from Google

*Table 8.* Overall performance on real-world noisy CIFAR-N datasets. The highest and second-highest accuracies are marked with bold and underlined text. Methods without any markings indicate the results are taken directly from the original papers, while * indicate the results were reproduced by us.

| Method | CIFAR-10N | | | | | CIFAR-100N |
|---|---|---|---|---|---|---|
| | *Aggregate* | *Random 1* | *Random 2* | *Random 3* | *Worst* | *Fine* |
| CE (Standard) | $87.77 \pm 0.38$ | $85.02 \pm 0.65$ | $86.46 \pm 1.79$ | $85.16 \pm 0.61$ | $77.69 \pm 1.55$ | $55.50 \pm 0.66$ |
| GCE (Zhang & Sabuncu, NeurIPS, 2018) | $87.85 \pm 0.70$ | $87.61 \pm 0.28$ | $87.70 \pm 0.56$ | $87.58 \pm 0.29$ | $80.66 \pm 0.35$ | $56.73 \pm 0.30$ |
| Co-teaching (Han et al., NeurIPS, 2018) | $91.20 \pm 0.13$ | $90.33 \pm 0.13$ | $90.30 \pm 0.17$ | $90.15 \pm 0.18$ | $83.83 \pm 0.13$ | $58.73 \pm 0.26$ |
| DMI (Xu et al., NeurIPS, 2019) | $89.43 \pm 0.11$ | $87.27 \pm 0.33$ | $86.96 \pm 0.21$ | $87.11 \pm 0.39$ | $80.36 \pm 0.19$ | $50.54 \pm 0.41$ |
| Co-teaching+ (Yu et al., ICML, 2019) | $90.61 \pm 0.22$ | $89.70 \pm 0.27$ | $89.47 \pm 0.18$ | $89.54 \pm 0.22$ | $83.26 \pm 0.17$ | $57.88 \pm 0.24$ |
| RoG (Lee et al., ICML, 2019) | $91.35 \pm 0.07$ | $90.48 \pm 0.25$ | $90.76 \pm 0.18$ | $90.37 \pm 0.23$ | $84.99 \pm 0.11$ | $58.49 \pm 0.26$ |
| Peer Loss (Liu & Guo, ICML, 2020) | $90.75 \pm 0.25$ | $89.06 \pm 0.11$ | $88.76 \pm 0.19$ | $88.57 \pm 0.09$ | $82.00 \pm 0.60$ | $57.59 \pm 0.61$ |
| ELR+ (Liu et al., NeurIPS, 2020) | $94.83 \pm 0.10$ | $94.43 \pm 0.41$ | $94.20 \pm 0.24$ | $94.34 \pm 0.22$ | $91.09 \pm 1.60$ | $66.72 \pm 0.07$ |
| DivideMix (Li et al., ICLR, 2020) | $95.01 \pm 0.71$ | $95.16 \pm 0.19$ | $95.23 \pm 0.07$ | $95.21 \pm 0.14$ | $92.56 \pm 0.42$ | $71.13 \pm 0.48$ |
| CAL (Zhu et al., CVPR, 2021) | $91.97 \pm 0.32$ | $90.93 \pm 0.31$ | $90.75 \pm 0.30$ | $90.74 \pm 0.24$ | $85.36 \pm 0.16$ | $61.73 \pm 0.42$ |
| CORES2 (Cheng et al., ECCV, 2022) | $95.25 \pm 0.09$ | $94.45 \pm 0.14$ | $94.88 \pm 0.31$ | $94.74 \pm 0.03$ | $91.66 \pm 0.09$ | $55.72 \pm 0.42$ |
| ASL (Zhou et al., TPAMI, 2023) | $90.01 \pm 0.37$ | $88.68 \pm 0.28$ | $88.12 \pm 0.21$ | $88.81 \pm 0.71$ | $79.23 \pm 0.70$ | $58.17 \pm 0.73$ |
| Pi-DUAL (Wang et al., ICML, 2024) | — | — | — | — | $84.91 \pm 0.40$ | $64.22 \pm 0.35$ |
| SGN (Englesson & Azizpour, ICLR, 2024) | $92.06 \pm 0.12$ | $91.94 \pm 0.19$ | $91.69 \pm 0.22$ | $91.91 \pm 0.10$ | $86.67 \pm 0.42$ | $60.36 \pm 0.71$ |
| CSGN (Lin et al., NeurIPS, 2024) | $94.01 \pm 0.12$ | $95.87 \pm 0.08$ | $95.47 \pm 0.12$ | $95.45 \pm 0.09$ | $95.54 \pm 0.06$ | $71.99 \pm 0.08$ |
| JAL (Wang et al., ICCV, 2025) | $90.06 \pm 0.22$ | $88.71 \pm 0.30$ | — | — | $81.25 \pm 0.10$ | $59.38 \pm 0.24$ |
| PCSE (Luo et al., TPAMI, 2025) | $92.02 \pm 0.13$ | $91.19 \pm 0.21$ | $91.21 \pm 0.14$ | $91.13 \pm 0.04$ | $85.81 \pm 0.20$ | $59.75 \pm 0.47$ |
| Early-Cutting (Yuan et al., NeurIPS, 2025) | $92.77 \pm 0.21$ | $92.50 \pm 0.14$ | $92.65 \pm 0.11$ | $92.36 \pm 0.43$ | $87.43 \pm 0.13$ | $66.52 \pm 0.22$ |
| IDO (Zhang et al., NeurIPS, 2025)* | — | — | — | — | $92.49 \pm 0.09$ | $73.32 \pm 0.15$ |
| PSSCL (Zhang et al., PR, 2025) | $96.41 \pm 0.20$ | **$96.17 \pm 0.12$** | $96.21 \pm 0.09$ | **$96.49 \pm 0.16$** | $95.12 \pm 0.10$ | $72.00 \pm 0.48$ |
| CSC (Fan & Li, AAAI, 2025) | **$96.42 \pm 0.24$** | $96.00 \pm 0.27$ | $95.80 \pm 0.33$ | $96.10 \pm 0.26$ | $93.10 \pm 0.51$ | $65.50 \pm 0.41$ |
| JYP (Ours) | $96.19 \pm 0.16$ | $96.07 \pm 0.22$ | **$96.28 \pm 0.25$** | $96.33 \pm 0.26$ | **$95.67 \pm 0.11$** | **$73.87 \pm 0.30$** |
| SIMIFEAT+ViT (Zhu et al., ICML, 2022)* | — | — | — | — | $94.45 \pm 0.24$ | $74.26 \pm 0.34$ |
| EPL+ViT (Ko et al., ICLR, 2023)* | — | — | — | — | $94.57 \pm 0.25$ | $74.32 \pm 0.37$ |
| CLIPCleaner (Feng et al., MM, 2024)* | — | — | — | — | $95.35 \pm 0.32$ | $73.21 \pm 0.41$ |
| NoiseGPT (Wang et al., NeurIPS, 2024) | $97.66 \pm 0.18$ | $97.48 \pm 0.24$ | — | — | $96.90 \pm 0.36$ | $73.24 \pm 0.45$ |
| LRAD+ViT (Chen et al., NeurIPS, 2024)* | $96.65 \pm 0.19$ | $96.55 \pm 0.23$ | $96.50 \pm 0.19$ | $96.53 \pm 0.22$ | $95.71 \pm 0.21$ | $74.14 \pm 0.31$ |
| PCL+ViT (Li et al., ICML, 2025)* | $95.88 \pm 0.17$ | $95.49 \pm 0.13$ | $95.37 \pm 0.08$ | $95.41 \pm 0.24$ | $94.78 \pm 0.23$ | $73.16 \pm 0.35$ |
| JYP+ViT (Ours) | **$97.72 \pm 0.18$** | **$97.50 \pm 0.12$** | **$97.46 \pm 0.08$** | **$97.45 \pm 0.16$** | **$97.21 \pm 0.09$** | **$75.03 \pm 0.44$** |

and Flickr based on the ILSVRC12 category hierarchy. Following prior work (Li et al., 2023), we use the first 50 classes from the Google subset of WebVision for training. Performance is evaluated on the validation sets of both WebVision and ILSVRC12 to assess generalization under large-scale, weakly labeled conditions.

### D.2. Evaluation Metrics

In addition to standard classification accuracy, we also adopt sample-selection-related evaluation metrics to verify the effectiveness of HCI (see Fig. 3d). These metrics mainly include Clean Precision (CP), Noisy Precision (NP), Clean Recall (CR), Noisy Recall (NR), Clean F1 (CF1), and Noisy F1 (NF1).

Let $\mathcal{S}_c$ and $\mathcal{S}_n$ denote the ground-truth sets of clean and noisy samples in the training data, respectively, and let $\hat{\mathcal{S}}_c$ and $\hat{\mathcal{S}}_n$ denote the sets of samples selected as clean and noisy by the our method. The evaluation metrics are then defined as follows:

$$\mathrm{CP} = \frac{|\hat{\mathcal{S}}_c \cap \mathcal{S}_c|}{|\hat{\mathcal{S}}_c|}, \quad \mathrm{NP} = \frac{|\hat{\mathcal{S}}_n \cap \mathcal{S}_n|}{|\hat{\mathcal{S}}_n|}, \tag{43}$$

$$\mathrm{CR} = \frac{|\hat{\mathcal{S}}_c \cap \mathcal{S}_c|}{|\mathcal{S}_c|}, \quad \mathrm{NR} = \frac{|\hat{\mathcal{S}}_n \cap \mathcal{S}_n|}{|\mathcal{S}_n|}, \tag{44}$$

$$\mathrm{CF1} = \frac{2 \cdot \mathrm{CP} \cdot \mathrm{CR}}{\mathrm{CP} + \mathrm{CR}}, \quad \mathrm{NF1} = \frac{2 \cdot \mathrm{NP} \cdot \mathrm{NR}}{\mathrm{NP} + \mathrm{NR}}. \tag{45}$$

Specifically, CP and NP measure the precision of sample selection, indicating the proportions of truly clean and truly noisy samples among those selected as clean and noisy, respectively. In contrast, CR and NR measure the recall of sample selection, reflecting the proportions of ground-truth clean and noisy samples that are correctly identified by the method. Together, CF1 and NF1 characterize the quality of sample selection from complementary precision and recall perspectives for both clean and noisy samples.

*Table 9.* Overall performance on Animal-10N and Clothing1M datasets. The highest and second-highest accuracies are marked with bold and underlined text. Methods without any markings indicate the results are taken directly from the original papers, while * indicate the results were reproduced by us.

| Method | Animal10N | Method | Clothing1M |
|---|---|---|---|
| CE (Standard) | 79.40 | CE (Standard) | 68.94 |
| Mixup (Zhang et al., ICLR, 2018) | 82.70 | Co-teaching (Han et al., NeurIPS, 2018) | 69.21 |
| GCE (Zhang & Sabuncu, NeurIPS, 2018) | 81.17 | JoCoR (Wei et al., CVPR, 2020) | 70.30 |
| SELFIE (Song et al., ICML, 2019) | 81.89 | DMI (Xu et al., NeurIPS, 2019) | 72.46 |
| Co-teaching (Han et al., NeurIPS, 2018) | 84.86 | DivideMix (Li et al., ICLR, 2020) | 74.45 |
| RoG (Lee et al., ICML, 2019) | 85.04 | ELR+ (Liu et al., NeurIPS, 2020) | 74.39 |
| DivideMix (Li et al., ICLR, 2020) | 84.50 | GJS (Englesson & Azizpour, NeurIPS, 2021) | 71.64 |
| PLC (Wei et al., ICLR, 2021) | 83.40 | AugDesc (Nishi et al., CVPR, 2021) | 74.33 |
| GJS (Englesson & Azizpour, NeurIPS, 2021) | 84.20 | CC (Zhao et al., ECCV, 2022) | 74.40 |
| NCT (Chen et al., TNNLS, 2022) | 84.10 | UNICON (Karim et al., CVPR, 2022) | 74.98 |
| SSR (Feng et al., BMVC, 2022) | 88.50 | OT-Filter (Feng et al., CVPR, 2023) | 74.51 |
| CMW-Net (Shu et al., TPAMI, 2023) | 84.70 | DISC (Li et al., CVPR, 2023) | 74.79 |
| OT-Filter (Feng et al., CVPR, 2023) | 85.50 | K-SPR (Wang et al., TPAMI, 2024) | 75.20 |
| DISC (Li et al., CVPR, 2023) | 87.10 | RML (Li et al., AAAI, 2024) | 75.14 |
| HMW (Zhang et al., AAAI, 2024) | 86.50 | SGN (Englesson & Azizpour, ICLR, 2024) | 73.87 |
| LRAD (Chen et al., NeurIPS, 2024) | 88.60 | LRAD (Chen et al., NeurIPS, 2024) | 75.70 |
| CLIPCleaner (Feng et al., MM, 2024) | 88.90 | PCSE (Luo et al., TPAMI, 2025) | 71.37 |
| PCSE (Luo et al., TPAMI, 2025) | 85.48 | PLReMix (Liu et al., WACV, 2025) | 74.85 |
| PCL (Li et al., ICML, 2025)* | 89.07 | IDO (Zhang et al., NeurIPS, 2025) | 74.85 |
| DLD (Hou et al., CVPR, 2025) | 89.40 | DULC (Xu et al., AAAI, 2025) | 75.09 |
| CSC (Fan & Li, AAAI, 2025) | 89.70 | PCL (Li et al., ICML, 2025)* | 74.51 |
| JYP (Ours) | **90.28** $\pm$ 0.12 | JYP (Ours) | **75.77** $\pm$ 0.07 |

## D.3. Implementation Details

In our experiments, we adopt ResNet34 and ResNet50 as trainable encoders for the CIFAR datasets and real-world datasets, respectively, as illustrated by the blue ResNet blocks in Fig. 7. For CIFAR experiments, all projection layers are set to a feature dimension of 512, while a larger dimension of 1024 is used for real-world datasets. All models are trained for 200 epochs using the Adam optimizer with a batch size of 256. The initial learning rate is set to 0.001 and is adjusted using an adaptive schedule consisting of a warm-up phase followed by a half-cycle cosine decay. Following FixMatch (Sohn et al., 2020), we apply both weak and strong data augmentations to generate two different views during the sample repartitioning phase. Specifically, weak augmentation includes resizing, random horizontal flipping, and random cropping, while strong augmentation adopts RandAugment (Cubuk et al., 2020) with $n = 2$ and $m = 10$, where two augmentation operations are randomly selected from a predefined pool and are applied with an intensity of 10%. All comparison results are either directly taken from the original papers or are reproduced by us using their default settings. All experiments are conducted on eight NVIDIA RTX 4090 GPUs, and all reported results are averaged over five runs with different random seeds.

## E. More Comprehensive Experimental Results and Analysis

### E.1. Results on Synthetic Noisy Datasets

**Results under CCN.** Under CCN settings, JYP consistently achieves strong performance across different noise rates and corruption patterns under both ResNet and ViT backbones. As shown in Table 6, in the ResNet setting, JYP attains the best or second-best results on CIFAR-10 under moderate to severe noise, and shows clear advantages under challenging conditions such as Sym 80% and Asym 40%. On CIFAR-100, JYP particularly excels under high noise levels, achieving the best performance under Sym 80%, which highlights its robustness when label corruption is severe and class ambiguity is high. In the ViT setting, JYP further strengthens its advantage, achieving the best results across all CCN settings on both CIFAR-10 and CIFAR-100, with especially notable gains under high noise rates. These results indicate that JYP effectively mitigates CCN and benefits from stronger feature representations.

**Results under IDN.** IDN poses a more challenging scenario due to its sample-specific and heterogeneous nature. As shown in Table 7, in the ResNet setting, JYP consistently outperforms or matches the strongest baselines on CIFAR-10 under moderate and high noise rates, and maintains competitive performance on CIFAR-100, particularly under severe

*Table 10.* Overall performance on WebVision and ILSVRC12 datasets. The highest and second-highest accuracies are marked with bold and underlined text. Methods without any markings are taken directly from the original papers, while * indicate the results were reproduced by us.

| Method | WebVision | | ILSVRC12 | |
|---|---|---|---|---|
| | top1 | top5 | top1 | top5 |
| F-correction (Patrini et al., CVPR, 2017) | 61.12 | 82.68 | 57.36 | 82.36 |
| Decoupling (Malach & Shalev-Shwartz, NeurIPS, 2017) | 62.54 | 84.74 | 58.26 | 82.26 |
| MentorNet (Jiang et al., ICML, 2018) | 63.00 | 81.40 | 57.80 | 79.92 |
| Co-teaching (Han et al., NeurIPS, 2018) | 63.58 | 85.20 | 61.48 | 84.70 |
| D2L (Ma et al., ICML, 2019) | 62.68 | 84.00 | 57.80 | 81.36 |
| MentorMix (Jiang et al., ICML, 2020) | 76.00 | 90.20 | 72.90 | 91.10 |
| DivideMix (Li et al., ICLR, 2020) | 77.32 | 91.64 | 75.20 | 90.84 |
| ELR+ (Liu et al., NeurIPS 2020) | 77.78 | 91.68 | 70.29 | 89.76 |
| GJS (Englesson & Azizpour, NeurIPS, 2021) | 77.99 | 90.62 | 74.33 | 90.33 |
| UNICON (Karim et al., CVPR, 2022) | 77.60 | 93.44 | 75.29 | 93.72 |
| CC (Zhao et al., ECCV, 2022) | 79.36 | 93.64 | 76.08 | 93.86 |
| CMW-Net (Shu et al., TPAMI, 2023) | 80.44 | 93.36 | 77.36 | 93.48 |
| DISC (Li et al., CVPR, 2023) | 80.28 | 92.28 | 77.44 | 92.28 |
| K-SPR (Wang et al., TPAMI, 2024) | 77.96 | 92.28 | 75.20 | 92.88 |
| CLIPCleaner (Feng et al., MM, 2024) | 81.56 | 93.26 | 77.80 | 92.08 |
| HMW (Zhang et al., AAAI, 2024) | 78.04 | 93.08 | 71.88 | 92.20 |
| RML (Li et al., AAAI, 2024) | 81.34 | — | 77.38 | — |
| SGN (Englesson & Azizpour, ICLR, 2024) | 76.12 | 90.74 | 72.72 | 90.35 |
| CSGN (Lin et al., NeurIPS, 2024) | 79.84 | 93.52 | 76.56 | 93.76 |
| LRAD (Chen et al., NeurIPS, 2024) | 84.16 | — | 82.56 | — |
| PCSE (Luo et al., TPAMI, 2025) | 70.48 | — | 67.72 | — |
| PLReMix (Liu et al., WACV, 2025) | 81.49 | 93.79 | 77.75 | 93.08 |
| Early-Cutting (Yuan et al., NeurIPS, 2025) | 73.81 | — | 71.20 | — |
| DLD (Hou et al., CVPR, 2025) | 84.51 | — | 83.74 | — |
| DULC (Xu et al., AAAI, 2025) | 79.90 | 93.70 | 76.90 | 93.90 |
| CSC (Fan & Li, AAAI, 2025) | 82.30 | 94.60 | 78.20 | 94.90 |
| DCD (Mu et al., ICCV, 2025) | 83.29 | 94.51 | 78.54 | 94.17 |
| PCL (Li et al., ICML, 2025)* | 84.20 | 95.28 | 82.42 | 94.39 |
| JYP One-step | 82.72 | 96.28 | 84.20 | 96.40 |
| JYP (Ours) | **84.62** | **97.60** | **85.08** | **98.20** |

noise. When combined with ViT, JYP achieves clear and consistent improvements across all IDN settings on both datasets, outperforming prior methods by noticeable margins. This demonstrates that JYP is especially effective at handling complex, feature-dependent noise patterns, and that its advantages are amplified when paired with more expressive pre-trained models.

### E.2. Results on Real-world Noisy Datasets

As shown in Tables 8– 10, JYP demonstrates strong robustness and generalization across diverse real-world noisy-label benchmarks. On CIFAR-N (Table 8), JYP achieves consistently competitive performance under multiple human-annotation settings, and obtains the best results on the most challenging *Worst* noise mode of CIFAR-10N (95.67%) as well as the best accuracy on CIFAR-100N *Fine* (73.87%), indicating its effectiveness under instance-dependent and human-centric noise. On Animal-10N (Table 9), JYP reaches 90.28%, outperforming prior methods and showing clear advantages on web-collected data with inter-class ambiguity. On the large-scale Clothing1M dataset (Table 9), JYP attains 75.77%, slightly surpassing the strongest baseline (75.70%), which verifies its scalability under severe, real-world, text-derived label noise. Furthermore, on WebVision/ILSVRC12 (Table 10), JYP achieves the best performance on both datasets (84.62/97.60 top1/top5 on WebVision and 85.08/98.20 on ILSVRC12), demonstrating strong robustness under large-scale weak supervision and improved cross-dataset generalization. Overall, these results confirm that JYP consistently handles heterogeneous noise sources (human, web, and weak textual labels) and scales well from small benchmarks to large real-world datasets.

### E.3. More Ablation Studies

**Why $y$-prediction is robust while $\epsilon$-prediction fails in LDMs.** Our experimental results in Fig. 10 reveal a clear asymmetry between different prediction parameterizations. In particular, $y$-prediction consistently achieves reasonable classification

performance across all combinations of prediction spaces and loss functions, whereas $\epsilon$-prediction exhibits catastrophic failure when paired with losses defined in the label space, with accuracy dropping to near-random levels.

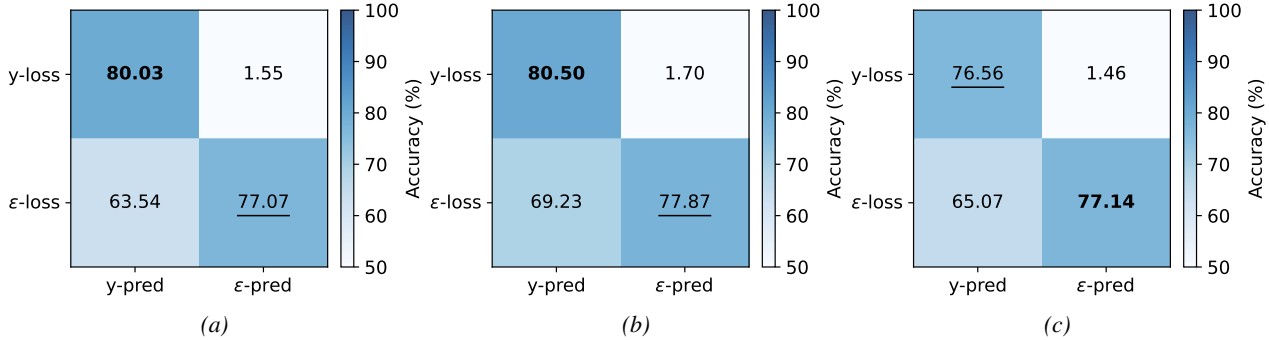

*Figure 10.* Classification performance of all combinations of prediction paradigms and loss functions (see Table 1) on the CIFAR-100 dataset under different noise settings. (a) CIFAR-100 w/ 50% Sym. (b) CIFAR-100 w/ 40% IDN (c) CIFAR-100N w/ *Fine* human noisy annotation.

We attribute this behavior to fundamental differences in numerical conditioning and task alignment. In LDMs, the clean target $\mathbf{y}_0$ lies in a bounded probability simplex and directly represents semantic class information. Predicting $\mathbf{y}_0$ therefore constitutes a well-conditioned and task-aligned objective: the network outputs are naturally constrained, and the supervision signal acts directly in the semantic space of interest. As a result, *y*-prediction remains stable even when optimized with losses defined in alternative spaces.

In contrast, $\epsilon$-prediction requires recovering $\mathbf{y}_0$ through an inverse process by Eq. 6, whose conditioning deteriorates at high noise levels where $\alpha_t$ becomes small. Errors in $\hat{\epsilon}$ are consequently amplified during the reconstruction of $\hat{\mathbf{y}}_0$, leading to unstable gradients. This issue is further exacerbated when employing $\boldsymbol{y}$-space losses such as CE, which impose sharp decision boundaries in the label space. Small reconstruction errors can therefore induce disproportionately large changes in the loss and its gradients, resulting in gradient saturation or explosion. The resulting mismatch between the prediction space ($\epsilon$) and the loss space ($\boldsymbol{y}$) drives the optimization toward degenerate solutions, explaining the observed catastrophic failures.

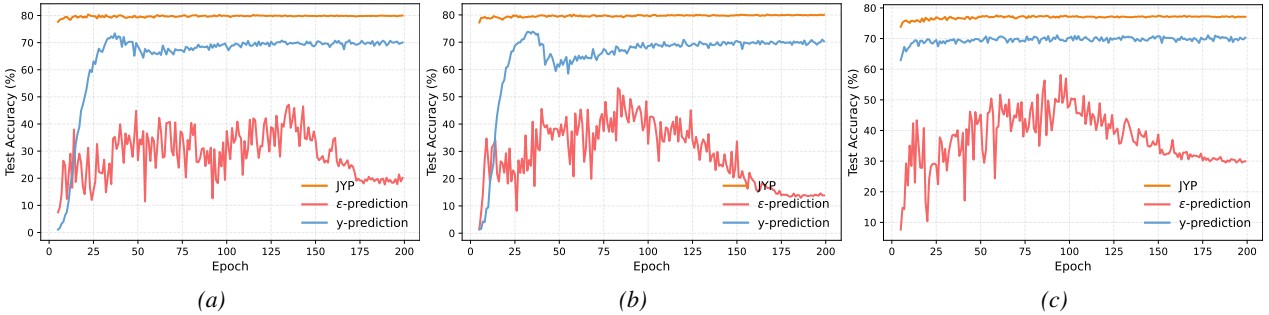

*Figure 11.* One-step inference performance on noisy CIFAR-100 using different LDM training methods. (a) CIFAR-100 w/ 50% Sym. (b) CIFAR-100 w/ 40% IDN. (c) CIFAR-100N w/ *Fine* human noisy annotation.

**JYP one-step inference performance and analysis.** Fig. 11 and Fig. 12 further demonstrate that LDMs trained with $\boldsymbol{y}$-prediction consistently achieve strong and stable one-step inference performance across all noisy CIFAR-100 settings, whereas models trained with $\epsilon$-prediction exhibit severe instability and eventual collapse. From the perspective of the *manifold hypothesis*, this discrepancy can be understood by examining what structure each prediction parameterization is encouraged to learn. In label diffusion, clean labels $\mathbf{y}_0$ reside on a low-dimensional, semantically meaningful manifold corresponding to the probability simplex of class membership. Direct $\boldsymbol{y}$-prediction therefore trains the network to learn the intrinsic *label manifold* and a contractive mapping that projects corrupted labels $\mathbf{y}_t$ back onto this manifold. As shown in Fig 12b, even after a single denoising step, predictions produced by JYP rapidly collapse from a diffuse, noise-dominated distribution into well-separated class-aligned clusters that closely match the ground-truth class structure.

Beyond synthetic benchmarks, Table 10 shows that JYP remains highly one-step performance on real-world noisy datasets. Notably, with *only one-step inference*, JYP achieves $82.72\%$ / $96.28\%$ top-1/top-5 accuracy on WebVision and $84.20\%$ /

96.40% on ILSVRC2012, outperforming or matching most prior SOTA methods. This indicates that JYP's gains do not depend on prolonged denoising. With full inference, performance further improves, but the strong one-step results alone demonstrate the effectiveness of learning a direct label-manifold projection under real-world noise.

In contrast, $\epsilon$-prediction trains the network to regress noise variables that are high-dimensional, unstructured, and largely independent of semantic class identity. This corresponds to learning a *noise manifold* that is both redundant and poorly aligned with the task-relevant label manifold. Recovering $\hat{\mathbf{y}}_0$ then requires an ill-conditioned inversion of the diffusion process, which amplifies prediction errors and produces outputs that may lie far off the label manifold, particularly at high noise levels. Consequently, while iterative sampling may partially correct these deviations over multiple steps, the learned denoiser does not implement a direct projection onto the label manifold, leading to brittle and unreliable one-step predictions. Consistent with this interpretation, the t-SNE visualizations (Fig. 12a) show that one-step $\epsilon$-prediction often fails to recover meaningful labels from the Gaussian initialization, leading to near-random predicted hard labels (colors) that mismatch the ground-truth classes (marker shapes). Notably, such mismatches persist for many samples even after multiple denoising iterations. This analysis highlights that, in LDMs, learning the target label manifold directly is crucial for enabling efficient and robust single-step inference.

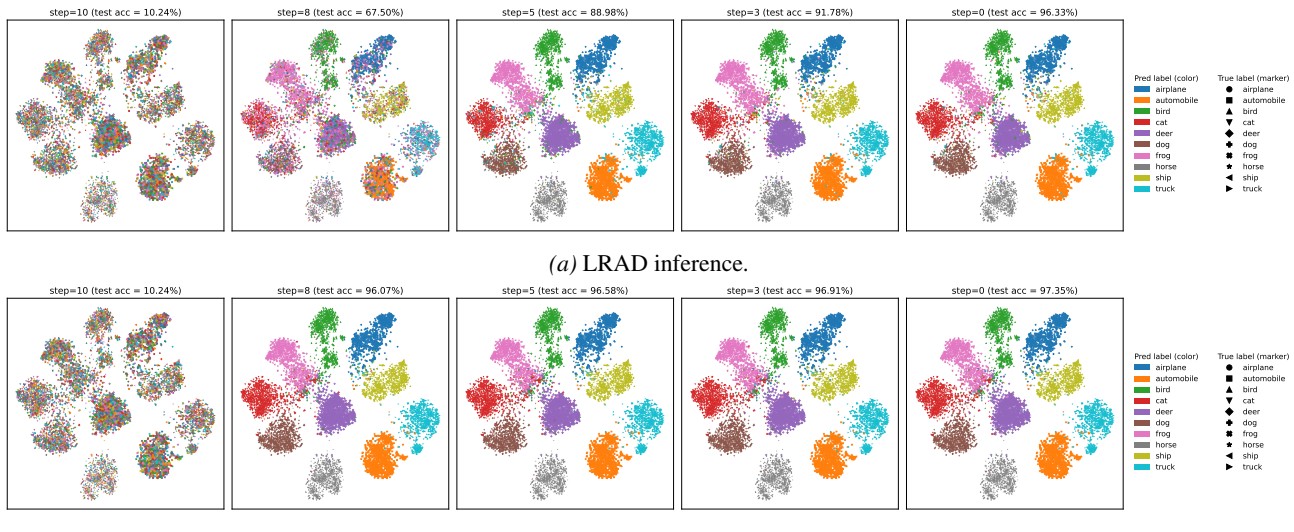

*(a)* LRAD inference.

*(b)* JYP inference.

*Figure 12.* Comparison of inference processes for different LDMs trained on the CIFAR-10 dataset w/ 50% Sym. The visualization is obtained by applying t-SNE (Maaten & Hinton, 2008) to the ViT feature embeddings of the test samples. Each data point represents one test image; marker shapes indicate the ground-truth classes, while colors correspond to the predicted hard labels at different inference steps. The titles of each subplot report the corresponding inference step and its test accuracy.

**Sensitivity analysis of the hyperparameter $\beta$.** In addition to network parameters, the proposed JYP introduces a single hyperparameter $\beta$ to control the relative learning weights of strong and weak views. We conduct a sensitivity analysis of $\beta$ on noisy CIFAR datasets, with results summarized in Fig. 13. Overall, the model performance is not highly sensitive to the choice of $\beta$: variations induced by changing $\beta$ are consistently smaller than those caused by different noise ratios or noise patterns. Across CIFAR-10/100 under both Sym. and IDN, the best performance is typically achieved within a broad range of $\beta \in [0.2, 0.4]$. Within this interval, the test accuracy remains relatively stable, indicating good robustness to hyperparameter selection and consistent generalization behavior. Under more severe or complex noise conditions (e.g., Sym. 80% or IDN 60%/80%), slightly larger values of $\beta$ tend to maintain or marginally improve performance, which aligns with the intuition that stronger suppression of the weak view helps mitigate noise interference. However, when $\beta$ becomes too large (e.g., close to 0.8), performance degradation is observed across most settings, suggesting that overly down-weighting the weak view may discard useful complementary information and harm generalization.

Based on these observations, we adopt $\beta = 0.2$ as the default setting in all experiments, which provides consistently strong performance across datasets and noise configurations.

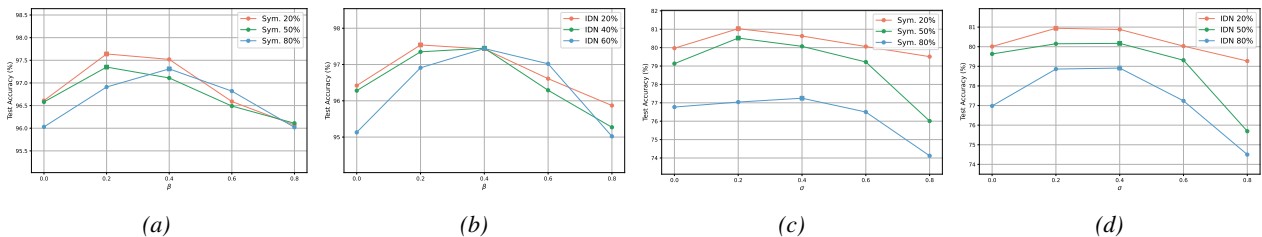

*Figure 13.* Sensitivity of model performance to the hyperparameter $\beta$ under different noise patterns on CIFAR. (a) CIFAR-10 w/ Sym. (b) CIFAR-10 w/ IDN (c) CIFAR-100 w/ Sym. (d) CIFAR-100 w/ IDN.

# F. Limitations and Future Work.

While the proposed method demonstrates strong robustness and consistent improvements across a wide range of noisy label settings, several limitations remain. First, our current study focuses on classification tasks with categorical labels, where the label space admits a well-structured and low-dimensional simplex geometry. Although this setting naturally benefits label diffusion and $\boldsymbol{y}$-prediction, extending the proposed framework to more complex structured outputs (e.g., multi-label, hierarchical, or continuous targets) remains an open question. Second, our approach relies on a predefined noise schedule and diffusion formulation; exploring adaptive or data-dependent noise schedules may further improve robustness, particularly under extremely high or heterogeneous noise conditions. Finally, while one-step inference is empirically effective in our experiments, a more rigorous theoretical understanding of when and why single-step denoising suffices remains lacking. We believe future work could address these limitations by extending label diffusion to richer output spaces, jointly learning diffusion dynamics with task objectives, and developing theoretical analyses that connect manifold structure, parameterization choice, and inference efficiency.

# G. Code Release and Reproducibility

The code and related implementation scripts for our proposed JYP method are publicly available at: `https://github.com/SenyuHou/JYP`. The released repository contains the main components required to reproduce the experimental results reported in this paper, including model architectures, diffusion training and inference procedures, HCI-based sample selection strategies, and evaluation scripts for synthetic and real-world noisy-label benchmarks.

