# OpenReview forum: "Just Y-Prediction: Enabling Historical Cumulative Inconsistency in Label Diffusion for Learning with Noisy Label"
_ICML.cc/2026/Conference — ICML 2026 regular_

### Official Review · Reviewer_Xsxs · 2026-03-10

**Soundness:** 3
**Presentation:** 3
**Significance:** 3
**Originality:** 3
**Overall Recommendation:** 4
**Confidence:** 4

**Summary:**

This paper addresses the limitations of existing label diffusion models (LDMs) in learning from labeled noise. The $\epsilon$-prediction paradigm relied upon by traditional LDMs lacks explicit semantic guidance due to the absence of clear category semantics in Gaussian noise, thereby constraining the model's reasoning and optimization capabilities in complex noisy environments. To address this, this paper introduces the y-prediction mechanism, which models directly on the label manifold to obtain explicit semantic guidance. Theoretical analysis demonstrates that y-prediction is equivalent to traditional $\epsilon$-prediction in terms of optimal solutions, ensuring mathematical rigor.

**Compliance With Llm Reviewing Policy:**

Affirmed.

**Key Questions For Authors:**

I hope the author can address the issues of theoretical ambiguity and inconsistency in the Weakness section.

**Limitations:**

yes

**Strengths And Weaknesses:**

Strengths:

The proposal of y-prediction is well-motivated, addressing the semantic ambiguity of the epsilon-prediction paradigm in noisy environments. The theoretical proof regarding the equivalence of optimal solutions is rigorous and provides strong justification for the proposed method. The introduction of Historical Cumulative Inconsistency is a principled approach to mitigate the instability of instantaneous predictions. By leveraging long-term generative behavior, the method achieves more reliable sample selection. The paper presents a thorough empirical analysis, including extensive ablation studies that clearly demonstrate the contribution of each component (JYP, HCI, and dual-view constraints). The consistent SOTA performance across multiple benchmarks reinforces the practical value of the work.

Weakness:

Theoretical Ambiguity and Inconsistency: Contradiction in mean function parametrization: There is a critical logical gap between Eq. (3) and Eq. (7). In Eq. (3), the authors defined the mean $\mu$ using the input feature $\mathbf{x}$ in the position where the ground-truth label $\mathbf{y}_0$ should reside. This suggests a confusion between 'conditioning guidance' and the 'diffusion object'.

Undefined Notation $x_{\theta}$: In Eq. (7), the term $x_{\theta}$ is introduced without definition. If the author intends this to be $y_\theta$ (predicting the label), it confirms that the derivation in Eq. (3) is fundamentally flawed. If it truly means predicting features $\mathbf{x}$, it contradicts the stated goal of y-prediction.

Lack of Formal Rigor: The manuscript consistently confuses scalars and vectors (e.g., subtracting scalar $\tilde{y}_i$ from vector $\mathbf{p}_i^{(t)}$) .

---

> ### Author Rebuttal · Authors · 2026-03-29
>
> **W1&W2**: Theoretical Ambiguity and Inconsistency: Contradiction in mean function parametrization: There is a critical logical gap between Eq. (3) and Eq. (7). In Eq. (3), the authors defined the mean $\mu$ using the input feature $x$ in the position where the ground-truth label $y_0$ should reside. This suggests a confusion between 'conditioning guidance' and the 'diffusion object'.
> Undefined Notation $x_\theta$: In Eq. (7), the term $x_\theta$ is introduced without definition. If the author intends this to be $y_\theta$(predicting the label), it confirms that the derivation in Eq. (3) is fundamentally flawed. If it truly means predicting features $x$, it contradicts the stated goal of y-prediction.
>
> **A1**: Thank you for pointing out the error in the use of symbols in Eq. (7). After checking, we found that it should be $y_\theta$ instead of $x_\theta$. Therefore, the incorrect formula of Eq. (7) was:
>
> $
> \mu_\theta(y_t,t) = \frac{A}{B} x_\theta(y_t,t) + \frac{C}{B} y_t,
> $
>
> and **the correct formula is**:
>
> $
> \mu_\theta(y_t,t) = \frac{A}{B} y_\theta(y_t,t) + \frac{C}{B} y_t.
> $
>
> which indicates that the training and prediction targets of the diffusion model are always the label $y$ rather than the feature $x$. **The feature $x$ should only serve as a control condition, and thus it should be included as a variable within the prediction function rather than being a predicted value**. We apologize for any confusion caused by the incorrect symbol usage. To ensure the rigor of the paper, we will further review and correct the equations and symbols throughout the text and make adjustments in the final version.
>
> **W3**: Lack of Formal Rigor: The manuscript consistently confuses scalars and vectors (e.g., subtracting scalar y from vector p) .
>
> **A2**: Thanks for your feedback regarding the mix of scalars and vectors. In fact, the overall method introduction does not involve scalars. For instance, although **the label $y_i$ is a numerical scalar, it is converted into a vector through one-hot encoding during the experimental process for computation purposes**. Therefore, we have decided to standardize the variables as vectors throughout the paper to ensure more consistent and clear symbol usage. We assure that these revisions will be included in the final version to maintain the theoretical rigor of the paper.

---

> > ### Author Rebuttal · Reviewer_Xsxs · 2026-04-03
> >
> > I keep my score unchanged.

---

> > > ### Author Response · Authors · 2026-04-03
> > >
> > > Thank you again for your positive evaluation and for the discussion on theoretical rigor. We will conduct a thorough review of the entire manuscript in the final version to ensure the accuracy and consistency of all formulas.

---

### Official Review · Reviewer_dc34 · 2026-03-10

**Soundness:** 3
**Presentation:** 3
**Significance:** 3
**Originality:** 3
**Overall Recommendation:** 4
**Confidence:** 3

**Summary:**

The authors address Learning with Noisy Labels (LNL) through the lens of Label Diffusion Models (LDMs), identifying fundamental limitations in the standard ε -prediction paradigm: lack of explicit class semantics during training, slow convergence, and inability to perform effective one-step inference.

**Compliance With Llm Reviewing Policy:**

Affirmed.

**Final Justification:**

The rebuttal addressed my main concerns and strengthened confidence in the paper’s empirical support. The final recommendation is 4.

**Key Questions For Authors:**

1. The Historical Cumulative Inconsistency mechanism is empirically effective for sample selection, but what theoretical guarantees can be provided regarding its ability to separate clean and noisy samples? For instance, under what conditions on the noise rate or data distribution does the GMM fitting on HCI achieve consistent sample identification? How does the time-dependent weighting $\phi_t$ affect the consistency of the selection?
2. Your method relies on frozen pretrained feature extractors ($f_p$) to bridge the dimensional gap between inputs and labels. How sensitive is JYP to the quality or architecture of these pretrained features? Have you evaluated the method with randomly initialized or simpler feature extractors, and would the advantages of JYP persist in regimes where powerful pretrained representations are unavailable?
3. While JYP enables efficient one-step inference, the training procedure involves dual-view augmentation, HCI accumulation, and GMM-based sample partitioning. What is the computational overhead of these additional components compared to standard LDM training or discriminative baselines like DivideMix? Is there a scenario where the training costs outweigh the inference benefits?

**Limitations:**

yes

**Strengths And Weaknesses:**

Strengths:
1. The theoretical foundation establishing the equivalence between prediction paradigms (Proposition 3.1) is rigorous and well-supported by the least-squares regression analysis in Appendix A.
2. The manuscript is generally well-organized with clear motivation illustrated in Figure 1. The distinction between $\varepsilon$-prediction (operating on high-dimensional noise manifolds) and $y$-prediction (operating directly on the label manifold) is intuitively explained. The experimental section is comprehensive, covering multiple noise types, datasets, and backbone architectures.

Weaknesses:
1. The HCI sample selection mechanism, while empirically effective, lacks theoretical guarantees regarding its consistency or sample complexity bounds for separating clean and noisy instances.
2. The paper would benefit from a deeper discussion of the optimization landscape differences between the two paradigms—specifically, why $y$-prediction avoids the numerical conditioning issues that plague $\varepsilon$-prediction when combined with cross-entropy losses (as noted in Appendix E.3). Some sections, particularly the description of the dual-view training procedure, are dense and could be clarified with additional algorithmic schematics or simplified pseudocode.
3. The core idea of $y$-prediction itself is not new to the diffusion community (having appeared in image generation contexts), so the primary novelty lies in the application to LNL and the specific HCI-based robust optimization framework.

---

> ### Author Rebuttal · Authors · 2026-03-30
>
> **W1&Q1: What theoretical guarantees for HCI to separate samples? How does time-dependent weighting affect selection?**
>
> **A1:** HCI aggregates historical information to build metrics for sample noise levels. It is inspired by recent works like DISC (Li et al. CVPR, 2023) and IDO (Zhang et al. , NeurIP, 2025), both of which use historical information for sample selection. From a theoretical perspective, Xia et al. (Sample Selection with Uncertainty of Losses for Learning with Noisy Labels, ICLR, 2022.) showed that **using interval estimation with historical training information reduces sample selection bias caused by noisy labels, providing guarantees on the statistical reliability of cumulative estimates**. We plan to explore the advantages of HCI over point estimation from a generalization error perspective in future work.
>
> Additionally, time-dependent weighting balances metric biases induced by varying time steps. Without it, sample selection accuracy drops by over 15%, mainly due to unfair comparisons between confusing samples. Thus, **time-dependent weighting is essential for effective HCI sample selection**.
>
> **W2: The paper needs more discussion on y-prediction's stability vs. ϵ-prediction and algorithmic diagrams for training.**
>
> **A2:** Firstly, y-prediction ensures stable optimization by directly predicting the label, while ϵ-prediction suffers from numerical instability due to amplified reconstruction errors at high noise levels. We will discuss these stability advantages in more detail in the final version. Secondly, the training process is detailed in Algorithm 2, and we will add pseudocode or diagrams in Appendix to improve clarity.
>
> **W3: What is the innovation of this work in applying y-prediction to LNL?**
>
> **A3:** Thanks for your discussion on the novelty of our work. **Our contribution lies in the novel application of y-prediction to the label diffusion paradigm, exploring its effectiveness in LNL problem for the first time**. Unlike the traditional x-prediction methods, **y-prediction faces challenges in efficiently utilizing the semantic information from generative models for sample selection and training**. This work provides a new generative learning framework for LNL and offers insights for the application and extension of diffusion models, making it academically valuable.
>
> **Q2: How sensitive is JYP to the quality or architecture of pretrained feature extractors?**
>
> **A4:** To assess JYP's dependency on pre-trained features, we used an unfinetuned ResNet model (with pretrained=True by default for classification) as the feature extractor. As shown in Table 1, even **without fine-tuning the pre-trained extractor, JYP maintains comparable performance to SOTA methods** (IDO, NIPS, 2025). LRAD (i.e., standard LDMs) shows a significant performance drop when the strong pre-trained model is removed, while JYP effectively selects clean and noisy samples through HCI, maintaining strong robustness and high performance even without powerful pre-trained features. Besides, using pre-trained models to enhance robustness is common in LNL tasks. Our method **makes more efficient use of limited pre-trained features**. As shown in the last three rows, JYP consistently outperforms existing methods (e.g., EPL, ICML, 2022) using the same pre-trained features, demonstrating its superior capability to leverage these features.
>
> **Table 1. Acc (%) comparison of different pre-trained features**
> |Method|Features|CIFAR-10|||CIFAR-100|||
> |-|-|-|-|-|-|-|-|
> |||50% Sym.|40% IDN|Human|50% Sym.|40% IDN|Human|
> |IDO|ResNet|$95.45$|93.78|92.49|$78.81$|$78.56$|73.32|
> |LRAD|ResNet|89.57|90.31|88.26|75.43|75.27|71.93|
> |JYP|ResNet|95.36|$95.18$|$94.67$|77.37|78.13|$73.69$|
> |EPL|ViT|96.10|95.90|94.57|76.50|77.80|74.32|
> |LRAD|ViT|96.34|96.75|95.71|77.07|77.87|74.14|
> |JYP|ViT|**97.35**|**97.45**|**97.22**|**80.52**|**80.15**|**75.03**|
>
> **Q3: What is the computational cost of JYP compared to standard LDM training or DivideMix?**
>
> **A5:** We conducted comparison experiments on the computational overhead in both training and inference. As shown in Table 2, **JYP significantly outperforms DivideMix in training time, as it only uses a single network model**. Compared to the standard LRAD, its time overhead is not significantly increased, indicating that **the dual-view and HCI partition technique do not introduce a heavy burden** on the model. These modules only slightly increase memory consumption, which remains within a reasonable range, without causing a situation where training time outweighs inference time. Detailed time overhead analysis is in our response to Reviewer #1 (cFo9) in A2.
>
> **Table 2. Training time per epoch (s) / GPU memory consumption (GB) of different methods on CIFAR-10, all methods use ResNet34 as backbone with batch size 256**
> |Method|JYP(Ours)|LRAD|Co-teaching|DISC|DivideMix|IDO|
> |-|-|-|-|-|-|-|
> |Time|24|18|20|36|54|201|
> |GPU|9.67|6.71|4.53|4.84|7.57|20.73|

---

> > ### Author Rebuttal · Reviewer_dc34 · 2026-04-02
> >
> > The additional clarification has addressed my earlier concerns, and I will maintain my positive score.

---

> > > ### Author Response · Authors · 2026-04-02
> > >
> > > Thank you for your recognition and positive feedback. We wish you success in your research and work.

---

### Official Review · Reviewer_aTxw · 2026-03-12

**Soundness:** 4
**Presentation:** 4
**Significance:** 3
**Originality:** 4
**Overall Recommendation:** 5
**Confidence:** 4

**Summary:**

This paper studies learning with noisy labels using label diffusion models. Instead of the standard ϵ-prediction objective, the authors propose Just Y-Prediction (JYP), which directly predicts the clean label  from a noisy label state. They further introduce Historical Cumulative Inconsistency (HCI) to separate clean, noisy, and hard samples, and combine it with dual-view sample selection and subset-specific robust training. Experiments on synthetic and real-world noisy-label benchmarks show strong performance, especially for one-step inference.

**Compliance With Llm Reviewing Policy:**

Affirmed.

**Final Justification:**

The rebuttal adequately addressed the main concerns and strengthened confidence in the paper’s empirical support, so the overall recommendation is raised from 4 to 5.

**Key Questions For Authors:**

1. Can the authors provide a stricter ablation that isolates the effect of y-prediction vs. ϵ-prediction under exactly the same training pipeline?
2. Can the authors report training and inference cost compared with strong discriminative baselines and prior diffusion-based methods?
3. How stable is HCI-based sample selection under class imbalance or long-tail noise?

**Limitations:**

Yes. The paper discusses limitations, including that the method is currently validated mainly for categorical labels and that the theory does not yet fully explain the strong practical effectiveness of one-step y-prediction. These limitations are acknowledged reasonably clearly.

**Strengths And Weaknesses:**

Strengths:The paper has a clear and original idea: directly predicting clean labels makes the diffusion objective better aligned with classification. The method is technically well motivated, and the paper provides both theoretical discussion and strong empirical results. The experiments are extensive, covering multiple noisy-label settings and large-scale real-world datasets. The one-step inference capability is also an appealing practical advantage.

Weaknesses:The main theoretical result only establishes equivalence at the population level, and does not fully explain why y-prediction is much easier to optimize in practice. Also, the final performance depends on several components beyond JYP itself, so the isolated contribution of the new parameterization is not completely clear. Finally, the method appears computationally heavier than standard discriminative noisy-label methods, while efficiency analysis is limited.

---

> ### Author Rebuttal · Authors · 2026-03-29
>
> **Q1: Can the authors provide a stricter ablation isolating the effect of y-prediction vs. ϵ-prediction with the same training pipeline?**
>
> **A1:** Thanks for your suggestion to further highlight the y-prediction. To more rigorously demonstrate the impact of these two prediction modes, we applied both y-prediction and ϵ-prediction on LRAD model, while strictly isolating other robust training strategies. The results (Table 1) show that, in a multi-step inference paradigm, y-prediction is competitive. **Notably, we don’t focus on y-prediction consistently outperforming ϵ-prediction, but rather on it maintaining comparable classification performance**. Because in the LNL problem, **y-prediction can provide crucial guidance for robust diffusion model optimization**, while **ϵ-prediction, with limited semantic information, cannot do so**. Moreover, y-prediction’s one-step performance significantly surpasses ϵ-prediction, which is one of its key advantages. We will include a detailed ablation analysis and discussion in the final version.
>
> **Table 1. Acc (%) comparison of different prediction paradigms on LRAD**
> |Method|50% Sym.|40% IDN|Human|
> |-|-|-|-|
> |ϵ|96.34|96.44|**95.71**|
> |y|**96.41**|**96.47**|95.59|
> |ϵ (One-step)|86.76|92.23|91.57|
> |y (One-step)|$94.85$|$95.26$|$93.44$|
>
> **Q2: Can the authors report training and inference cost?**
>
> **A2:** Thanks for your suggestion to include an analysis of the computational overhead. We conducted comparison experiments on the computational overhead in both training and inference:
>
> (i) As shown in Table 2, JYP requiring only 24s per epoch for training, significantly outperforming other methods (e.g., IDO requires 201s). Although added the dual-view and HCI partition modules, these modules are primarily used for sample selection and do not introduce significant overhead. Besides, JYP's GPU memory consumption is slightly higher, mainly due to the cache for dual-view and HCI, but it remains within a reasonable range and is considerably lower than the SOTA method IDO. Overall, **JYP offers clear advantages in training time and computational cost**, reducing resource consumption while maintaining accuracy.
>
> **Table 2. Training time per epoch (s) / GPU memory consumption (GB) of different methods on CIFAR-10, all methods use ResNet34 as backbone with batch size 256**
> |Method|JYP(Ours)|LRAD|Co-teaching|DISC|DivideMix|IDO|
> |-|-|-|-|-|-|-|
> |Time|24|18|20|36|54|201|
> |GPU|9.67|6.71|4.53|4.84|7.57|20.73|
>
> (ii) Table 3 presents the comparison of inference time for JYP, LRAD, and a general discriminative classification model (represented by ResNet) on the entire test set. **JYP maintains efficiency comparable to discriminative model in single-step inference and outperforms LRAD**, which relies on multi-step inference.
>
> **Table 3. Inference or classification time overhead (s) on different test datasets**
> |Dataset|JYP(Ours)|LRAD|ResNet|
> |-|-|-|-|
> |CIFAR|1|2|1|
> |Animal-10N|2|5|1|
> |WebVision|4|21|2|
> |ILSVRC12|5|27|3|
>
> We will include the results and discussion on computational overhead in the final version.
>
> **Q3: How stable is HCI-based sample selection under class imbalance or long-tail noise?**
>
> **A3:** Thanks for your question. Regarding the stability of the HCI-based GMM partition in handling class imbalance, **JYP supports sample selection by class rather than using all samples, effectively mitigating the issue of inaccurate sample selection due to class imbalance**. To evaluate the stability, we resampled a noisy training set and adjusted the number of samples per class according to the imbalance factor, i.e., for each class $c$, the number of samples is set as $N_c = N_c / \rho^{\frac{c-1}{C-1}}$, constructing various imbalance scenarios.
>
> The experimental results (Table 4) show that the JYP method exhibits strong robustness in class-imbalanced scenarios. When the imbalance factor $\rho=1$, the dataset is balanced. As $\rho$ increases, although the performance of all methods decreases, **JYP shows the smallest performance decay, demonstrating its stability in class-imbalanced situations**. In extreme imbalance scenarios (e.g., $\rho=10$), the scarcity of minority class samples may cause fitting bias in the HCI-based GMM, slightly reducing the sample selection accuracy and classification performance. However, JYP still outperforms methods that do not explicitly include class imbalance learning strategies.
>
> Notably, although **our method was not explicitly designed to address class imbalance, it still performs well in such scenarios**. In future work, we will consider incorporating class-aware training strategies or using upsampling techniques for minority classes to further improve its applicability in complex scenarios like class imbalance.
>
> **Table 4. Acc (%) comparison of CIFAR-10 w/ 20% Sym. under different class imbalance settings**
> |Method|$\rho=1$|$\rho=5$|$\rho=10$|
> |-|-|-|-|
> |DivideMix|96.10|93.91|74.78|
> |LRAD|96.18|94.31|89.27|
> |JYP(Ours)|**96.36**|**95.94**|**91.31**|

---

> > ### Author Rebuttal · Reviewer_aTxw · 2026-04-01
> >
> > Thank you for the detailed response. I have raised my score.

---

> > > ### Author Response · Authors · 2026-04-01
> > >
> > > Thank you for your reply and acknowledgment. We also wish you success in your research.

---

### Official Review · Reviewer_cFo9 · 2026-03-12

**Soundness:** 3
**Presentation:** 3
**Significance:** 2
**Originality:** 2
**Overall Recommendation:** 4
**Confidence:** 3

**Summary:**

The paper introduces a training paradigm "just y prediction" for label diffusion models in terms of noisy label learning. This method differs from the epsilon prediction, where it directly predicts the target label during the diffusion. This design is a principled change shown to be population-level equivalent to the conventional formulation. Specifically, it can help stabilize the optimization by associating it with the label manifold. This formulation helps enable efficient inference. Authors also propose a different inconsistency metric, which measures the prediction stability to categorize data according to a dual-view strategy. Evaluations on noisy benchmarks demonstrate that the proposed approach achieves competitive state-of-the-art performance compared with baselines.

**Compliance With Llm Reviewing Policy:**

Affirmed.

**Final Justification:**

The authors have addressed my concerns. Thus I maintain my support for this paper.

**Key Questions For Authors:**

Please see my comments above.

**Limitations:**

Please see my comments above.

**Strengths And Weaknesses:**

Strengths
Authors provide a principled and intuitive solution. They establish the population level optimal equivalence between y prediction and epsilon prediction.
By sidestepping the iterative denoising required to map high dimensional noise back to the label space, the proposed method can enable robust and accurate single step inference for generative classification.
The usage of the new inconsistency metric is interesting, and authors utilize stable semantic targets to tackle noisy samples.

Weaknesses
The authors effectively evaluate the model on standard multiclass image benchmarks. But the generalizability of the framework towards other modalities / structured label spaces can be additionally explored.
Can authors conduct an additional experiment with standard tabular dataset or multilabel text classification benchmarks to verify cross domain applicability?

Next, the dual view formulation incorporates sample partitioning with gaussian mixture model fitting as well as inconsistency accumulation. But this formulation can introduce additional moving parts, and these parts are not comprehensively discussed for computational costs.
Importantly, can authors add a computational result table? Here, we should compare the GPU memory consumption and training time per epoch against a standard baseline as well as a widely adopted discriminative model.

In addition, the proposed method relies on a fixed predefined Gaussian noise schedule. This can limit the flexibility, especially when handling highly extreme or heterogeneous instance dependent noise distributions.
Thus, is it possible to run a focused ablation study on the CIFAR dataset with instance dependent noise? We can compare the linear noise schedule with a different cosine noise schedule. Intuitively, this experiment checks if the formulation is sensitive to the forward transition kernel.

---

> ### Author Rebuttal · Authors · 2026-03-29
>
> **W1: Can the authors test on standard tabular or multilabel text benchmarks to verify cross-domain applicability?**
>
> **A1:** Thanks for your insightful discussion on the cross-domain applicability of our proposed method. We have validated the proposed JYP (which removes the ResNet module and retains only the linear layer) on several standard UCI tabular datasets, including text classification (CANE-9 and Spambase), speech (Isolet), image classification (Letter), and standard tabular classification datasets. We compared our method with traditional machine learning approaches (e.g., NB, SVM, RF, XGBoost) and similar LDM methods (LRAD). The results (Table 1) demonstrate clear advantages across different dataset types. On the CANE-9 dataset based on text frequency features, **JYP improved classification accuracy by 4.39%–10.88% over traditional machine learning models**. On standard tabular classification benchmarks, JYP showed improvements of 2.12%–5.30% compared to traditional models and an average improvement of 0.86% over LRAD. Overall, **our method excels across multiple data modalities, confirming its cross-domain adaptability and broad application potential**.
>
> **Table 1. Classification accuracy (%) on UCI tabular datasets**
> |Method/Dataset|CANE-9|Spambase|Isolet|Letter|Australian|Blood|Breast Cancer|Credit|Titanic|
> |-|-|-|-|-|-|-|-|-|-|
> |NB|83.56|91.37|77.56|64.41|84.78|69.00|96.72|75.22|74.80|
> |SVM|86.34|91.42|95.15|92.43|83.33|73.67|96.35|81.47|76.95|
> |RF|90.05|**94.35**|93.23|94.67|**86.96**|71.00|97.45|81.24|76.96|
> |XGBoost|87.27|94.19|92.95|94.20|86.75|72.33|97.45|81.09|76.96|
> |LRAD|92.59|93.91|95.87|94.61|86.23|77.00|97.08|81.82|78.09|
> |JYP(Ours)|**94.44**|94.18|**95.91**|**96.87**|86.23|**78.67**|**97.45**|**81.93**|**78.88**|
>
> **W2: Can the authors include a table comparing GPU memory and training time per epoch with a baseline and a common discriminative model?**
>
> **A2:** Thanks for your suggestion to include an analysis of the computational overhead. We conducted comparison experiments on the computational overhead in both training and inference:
>
> (i) Training Time and GPU Memory Consumption:
>
> Table 2 shows the training time and GPU memory consumption of the JYP method compared to other methods. JYP is relatively efficient in terms of training time, requiring only 24 seconds per epoch on the CIFAR dataset, which offers a clear advantage over other methods (e.g., IDO, which takes 201 seconds) in terms of computational cost. Furthermore, JYP retains a single-network architecture, although it adds some modules (e.g., dual-view and HCI partitioning), the dual-view is only used for sample selection, not for parameter updates during backpropagation. The GMM partitioning is based on one-dimensional HCI values and does not introduce significant overhead. Therefore, **the two key modules do not notably increase training time**. In addition, the GPU memory consumption of JYP is slightly higher than traditional methods, mainly due to the caching required for dual-view and HCI, which is within a reasonable range and comparable to other methods. It is still much lower than the memory overhead of the SOTA method IDO, making it acceptable at this level.
>
> **Table 2. Training time per epoch (s) / GPU memory consumption (GB) of different methods on CIFAR-10, all methods use ResNet34 as backbone with batch size 256**
> |Method|JYP(Ours)|LRAD|Co-teaching|DISC|DivideMix|IDO|
> |-|-|-|-|-|-|-|
> |Time|24|18|20|36|54|201|
> |GPU|9.67|6.71|4.53|4.84|7.57|20.73|
>
> (ii) Inference Time:
> Table 3 presents the comparison of inference time for JYP, LRAD, and a general discriminative classification model (represented by ResNet) on the entire test set. Specifically, the inference time on test datasets shows that **JYP maintains efficiency comparable to discriminative model in single-step inference and outperforms LRAD**, which relies on multi-step inference.
>
> **Table 3. Inference or classification time overhead (s) on different test datasets**
> |Dataset|JYP(Ours)|LRAD|ResNet|
> |-|-|-|-|
> |CIFAR|1|2|1|
> |Animal-10N|2|5|1|
> |WebVision|4|21|2|
> |ILSVRC12|5|27|3|
>
> Thank you again for your valuable suggestion. We will include the results and discussion on time overhead in the final version.
>
> **W3: Can authors compare the linear and cosine noise schedules to test sensitivity to the forward transition kernel?**
>
> **A3:** We conducted a sensitivity analysis experiment on the CIFAR-N dataset to compare the impact of different noise scheduling strategies in the forward transfer kernel on the method’s performance. The results (Table 4) show slight performance differences among the strategies, but all maintain high overall performance. Thus, the **JYP method is not highly sensitive to the choice of noise scheduling strategy**.
>
> **Table 4. Classification accuracy (%) on CIFAR-10N with different noise scheduling**
> |Schedule|linear|quad|jsd|sigmoid|cosine|cosine anneal|
> |-|-|-|-|-|-|-|
> |Acc|97.10|97.23|97.04|97.14|97.22|97.28|

---

> > ### Author Rebuttal · Reviewer_cFo9 · 2026-04-02
> >
> > The authors have addressed my concerns. Thus I maintain my support for this paper.

---

> > > ### Author Response · Authors · 2026-04-03
> > >
> > > Thank you for your support and recognition of our work. We wish you success in your research and work.

---

### Decision · Program_Chairs · 2026-04-30

**Decision:**

Accept (regular)

**Comment:**

After rebuttal, all reviewers show positive scores to this paper. Generally, the presented idea is interesting, and the authros also conducted intensive experiments to justify the effectiveness. Therefore, I recommend an acceptance.